# Effects of Calcium, Magnesium and Potassium on Sweet Basil Downy Mildew (*Peronospora belbahrii*)

**Yigal Elad** [1,*] **, Ziv Kleinman** [2] **, Ziv Nisan** [1,2,3] **, Dalia Rav-David** [1] **and Uri Yermiyahu** [4]

1   Department Plant Pathology and Weed Research, The Volcani Center, Agricultural Research Organization, Rishon LeZion 7534509, Israel; zivnisan@gmail.com (Z.N.); dalia@volcani.agri.gov.il (D.R.-D.)
2   Bikat HaYarden Research and Development, Tzevi Research Station, Bikat HaYarden 91906, Israel; ziv.kleinman@mail.huji.ac.il
3   The Robert H. Smith Faculty of Agriculture, Food and Environment, The Hebrew University of Jerusalem, Rehovot 7610001, Israel
4   Gilat Research Center, Agricultural Research Organization, Negev 85280, Israel; uri4@volcani.agri.gov.il
*   Correspondence: elady@volcani.agri.gov.il

**Abstract:** Downy mildew (*Peronospora belbahrii*) is a major disease of sweet basil (*Ocimum basilicum*). We examined the effects of potassium, calcium and magnesium, individually and in combination, on sweet basil downy mildew (SBDM) in potted plants and under commercial-greenhouse conditions over six growing seasons. An increased K concentration in the fertigation solution increased SBDM severity, whereas foliar-applied KCl and $K_2SO_4$ suppressed SBDM. The application of higher concentrations of those salts increased the K concentrations in the shoots and significantly alleviated SBDM. Increased concentrations of Ca or Mg in the fertigation solution decreased SBDM severity, as did foliar-applied $CaCl_2$. However, the combination of Ca and Mg did not have any synergistic effect. Foliar-applied $K_2SO_4$ provided better disease suppression than some of these treatments. The 3.3 mM Mg + fungicide treatment and the 5.0 mM Mg + fungicide treatment each provided synergistic disease control in one of two experiments. SBDM severity was significantly reduced by $MgCl_2$ and $MgSO_4$ (both 3.3 mM Mg), as compared with the basic Mg fertigation (1.6 mM), with $MgCl_2$ providing better control. The combined Mg salts + fungicide treatments reduced SBDM better than any of those treatments alone. These results demonstrate that macro-elements can contribute to SBDM control.

**Keywords:** calcium; downy mildew; fertigation; irrigation; macro-elements; magnesium; *Ocimum basilicum*; potassium; yield

## 1. Introduction

Macro-elements play an important role in plant nutrition and plant health. Among these, the cations $K^+$, $Ca^{2+}$ and $Mg^{2+}$ are particularly important [1,2]. $K^+$ is the most abundant cation in plant tissue; its concentration in the cytoplasm reaches 200 mM, and it accounts for up to 6% of dry plant weight [3]. $K^+$ is readily translocated in the phloem and xylem and moves through tissues and into cells via K channels [4]. $K^+$ helps to regulate the electrical charge in cells and control the acidity of the cytosol and chloroplasts and plays a role in enzymatic reactions [5]. $K^+$ also affects the process of photosynthesis, including the turgor pressure that opens and closes stomata, other osmoregulation, cell elongation, upregulation of enzyme expression and protein synthesis [3,6]. In sweet basil, K deficiency inhibits growth and leads to the development of thin stems, leaf-edge necrosis and slow-developing roots (Yigal Elad, unpublished data).

When present at optimal concentrations, K decreases plants' susceptibility to disease [1]. It increases the thickness of the walls of epidermal cells [4] and decreases the availability of sugar, amino acids and organic acids that are essential to the parasites' nutrition [3]. K regulates the plant's reaction to stress, including reactive oxygen species, as

well as jasmonic acid, ethylene and auxins [7,8]. Increased K levels are associated with high levels of the phenols rosmarinic acid and chicoric acid and increased anti-oxidative activity in sweet basil [9]. K is known to reduce the incidence diseases caused by the ascomycete pathogens *Botrytis cinerea* and *Sclerotinia sclerotiorum* [10,11]. It also effectively suppresses the severity of downy mildew (*Peronospora plantaginis*) in isabgol (*Plantago ovata*) [12].

The translocation of $Ca^{2+}$ in the xylem vessels of the plant is governed by transpiration during the day and by root pressure at night. Cations such as $Mg^{2+}$ and $K^+$ compete with $Ca^{2+}$ for absorption in the roots [13,14]. Ca accounts for between 0.1 and 5% of the plant's dry weight [4]. Ca mediates various activities in the plant tissues, including polar cell growth, cytoplasmic flow, mitosis and cytokinesis. It is a secondary messenger in signal pathways, an activator of enzymes and associates with Ca-binding proteins [14]. In sweet basil plants, Ca deficiency causes root degeneration, necrotic lesions on leaves and defoliation (Yigal Elad, unpublished data).

Ca is an important part of plant defense systems [15]. It is important for membrane function and stability and for preventing solutes like sugars and amino acids from leaking from the cytoplasm into the apoplast; these activities help to prevent disease. Ca is an important constituent of cell walls, and especially pectin, as it binds oligomers of pectin, and so helps to prevent pathogen penetration. It also acts against pathogens' cell wall-degrading enzymes [15,16]. Apart from those direct effects, Ca also plays a role in the expression of defense-related genes, the activity of pathogenesis-related proteins and hypersensitive reactions [17–19]. Ca increases plants' resistance to pathogens such as species of *Pythium*, *Sclerotinia*, *Botrytis* and *Fusarium* [20]. In sweet basil, Ca has been shown to reduce the incidence of *Botrytis cinerea* and *Sclerotinia sclerotiorum* [10,11]. Sprayed applications of Ca have been shown to reduce the severity of downy mildew (*Sclerospora graminicola*) in pearl millet (*Pennisetum glaucum*) [21].

The concentration of Mg in cells is 15 to 25 mM [22], and it is mostly found in the cells' organelles [23]. Mg is translocated in the phloem and accumulates in cell vacuoles. Its absorbance by the roots is especially affected by Ca, K and Mn [24]. It is involved in protein synthesis, DNA and RNA synthesis, phosphorylation, cell energy transformation, carbohydrate metabolism and movement [4]. Mg is also important for photosynthesis; about 20% of the Mg in plant cells is part of chlorophyll molecules [22]. In most plants, Mg deficiency is manifested as chlorosis and leaf necrosis [24]. Mg affects plant diseases, directly and indirectly, through its antagonistic interactions with other minerals (e.g., Ca, K and Mn) [25]. The severity of disease caused by *Fusarium oxysporum* f. sp. *conglutinans* in cotton (*Gossypium arboreum*) is reduced when Mg is available at optimal levels. A high concentration of Mg that interferes with Ca absorption increases the severity of bacterial speck (*Xanthomonas campestris* pv. *vesicatoria*) in tomato (*Solanum lycopersicum* [25]. Mg has also been shown to decrease the severity of tobacco downy mildew (*Peronospora tabacina*) [26].

Downy mildew (*Peronospora belbahrii*) is a major foliar disease of sweet basil (*Ocimum basilicum*) in Israel [27]. We examined the effects on sweet basil downy mildew (SBDM) of Ca, Mg, and K applied alone or in combination with each other through the irrigation solution or as a foliar spray. In this study, we first determined the optimal concentrations for applications of each of the mineral cations in potted plants and then examined the effects of a limited number of concentrations on SBDM under commercial conditions for six growing seasons (three years). We also examined the relationships between various nutritional elements in the sweet basil plants and SBDM severity.

## 2. Materials and Methods

### 2.1. Plants and Growing Conditions

Sweet basil (*Ocimum basilicum*) cv. Peri [28] was used for all of our experiments. The seedlings were grown in a commercial nursery (Shorashim, Mivtahim, Israel) and transplanted for the experiments 3 to 4 weeks after seeding in the nursery. The "Peri" cultivar is known to be susceptible to *Peronospora belbahrii* [27]. Plugs of sweet basil

seedlings were used, each containing three to five plants; hereafter, one plug is referred to as a "plant" as is common practice [11]. The experiments involving potted plants were performed at two sites in Israel: the Gilat Research Center in the northern Negev (Site A) and the Volcani Center in Rishon LeZion (Site B). Experiments were also carried out using plants grown in containers under commercial greenhouse conditions at the Tzvi Experimental Station in the Jordan Valley, Israel (Site C).

At Sites A and B, experiments were carried out in 2-L pots. At Site C, the experiments were carried out in 500-L containers. For the fertigation experiments, sweet basil plants were planted in the pots or containers filled with perlite (medium size, 1.2 mm, Agrifusia, Fertilizers & Chemicals Ltd., Haifa, Israel). For the experiments involving the foliar salt treatments, we used a potting mixture consisting of coconut fiber:tuff (unsorted to 8 mm; 7:3 vol.:vol.) The plants were irrigated to excess via a drip system two to four times a day, depending on the season, at a volume calibrated to lead to >30% water leaching. The daily irrigation volume was determined after analyzing the irrigation and drainage solutions once every 2 weeks, to prevent over-salinization or acidification of the root-zone solution. Plants in pots and containers were maintained according to the local extension service's recommendations. All pot experiments were irrigated with fresh water (electrical conductivity (EC) < 1.0 dS/m). The tested elements were applied with the irrigation water (fertigation) or as a foliar spray, as described below and summarized in Table 1.

**Table 1.** Experimental setup, factors tested, application methods and growing seasons [1].

| Site | Code | Growing Setting | Elements Tested | Additional Treatment | Application | Season |
|------|------|-----------------|-----------------|----------------------|-------------|--------|
| A | A-K-f | Pots | K (Cl) | | Fertigation (f) | All year |
| A | A-Ca-f | Pots | Ca (Cl) | | Fertigation | All year |
| A | A-Mg-f | Pots | Mg (Cl) | | Fertigation | All year |
| B | B1 [Ca-s, Mg-s, K-s] | Pots | K, Ca (Cl), Mg (Cl) | | Foliar (spray, "s") | All year |
| B | B2-K-s | Pots | K (Cl-$SO_4$) | | Foliar | All year |
| B | B3 [Ca-f, Mg-f, K-s] | Pots | K, Ca, Mg | | Fertigation: Ca + Mg Foliar K ($SO_4$) | All year |
| C | C1 a and b | Greenhouse | K, Ca, Mg | | Fertigation: Ca + Mg Foliar K ($SO_4$) | Spring (a) and Autumn (b) |
| C | C2 a and b | Greenhouse | Mg (Cl) | Fungicide spray | Fertigation | Autumn (a) and Spring (b) |
| C | C3 | Greenhouse | Mg (Cl-$SO_4$) | Fungicide spray | Fertigation | Autumn |

[1] Experiments A and B were repeated twice with 5–10 replicates each trial.

### 2.2. Pot Experiments

#### 2.2.1. Effects of Different Concentrations of K, Ca and Mg in the Fertigation Solution on SBDM (Experiments A)

Pot experiments were conducted in an unheated, polyethylene-covered greenhouse located at Site A. The aim of these experiments was to study the effects of different K, Ca and Mg concentrations in the fertigation solution ("f" treatments) on the development of SBDM in potted sweet basil plants. The sweet basil plants were planted in 2-L perlite-filled pots with one plant per pot, in 10 replicates, and each set of cation-concentration was repeated twice with 10 replicates each.

Nutrient solutions were prepared in 500-L containers containing all of the added nutrients. All of the plants were fertigated with 5-3-8 (N-$P_2O_5$-$K_2O$) fertilizer (Fertilizers and Chemical Compounds Ltd., Haifa, Israel) until the first shoot harvest. Later on, the effect of cation concentration was tested by tailoring the fertigation solution to each K, Ca and Mg concentration of interest, as described below. The concentrations of nutrients that were not part of the experiments and remained the same across all treatments were as follows: 5.7 mM N (90% $NO_3^-$-N and 10% $NH_4^+$-N), 0.35 mM P, 2.6 mM K (excluding Experiment A1-K below), 1.3 mM Ca (excluding Experiment A1-Ca below), 0.54 mM Mg (excluding Experiment A1-Mg below), 1.1 mM $SO_4^{-2}$, 0.023 mM B, 9.8 μM Fe, 4.9 μM Mn, 2.1 μM Zn, 0.31 μM Cu and 0.16 μM Mb. Solutions were prepared by dissolving $KH_2PO_4$,

$K_2SO_4$, $KNO_3$, $NH_4H_2PO_4$, $NaNO_3$, and $NH_4NO_3$ in water [29]. In Experiment A1-Ca, Ca was applied as $CaCl_2$ and, in Experiment A1-Mg, Mg was applied as $MgCl_2$.

When the plants reached a height of 45 to 50 cm, they were artificially inoculated with *P. belbahrii* as described below.

### 2.2.2. K Concentration in the Fertigation Solution (Experiment A-K-f)

The aim of these experiments was to study the effect of the K concentration in the fertigation solution on the development of SBDM in potted sweet basil plants. To characterize the response of sweet basil plants to different concentrations of K in the fertigation solutions, five K concentrations (0.5, 0.8, 1.3, 2.6 and 5.1 mM) were used while the concentrations of the other nutritional elements were kept constant. The EC in the different K-f treatments was 1.03–1.36 (dS)/m, and the pH of the fertigation solution was 6.99–7.42.

### 2.2.3. Ca Concentrations in the Fertigation Solution (Experiment A-Ca-f)

The aim of these experiments was to study the effect of Ca concentration on the development of SBDM in potted sweet basil plants. To characterize the response of sweet basil plants to different concentrations of Ca in the fertigation solutions, six Ca concentrations (0.50, 1.00, 1.75, 2.75, 4.00 and 6.00 mM) were used while the concentrations of the other nutritional were kept constant. The EC in the different Ca-f treatments was 0.99–2.50 dS/m and the pH of the fertigation solution was 6.43–7.12.

### 2.2.4. Mg Concentrations in the Fertigation Solution (Experiment A-Mg-f)

The aim of these experiments was to study the effect of the Mg concentration in the fertigation solution on the development of SBDM in potted sweet basil plants. To characterize the response of sweet basil plants to different concentrations of Mg, six Mg concentrations (0 (distilled water used), 0.4, 0.8, 1.6, 3.3 and 4.9 mM) were used while the concentrations of the other nutritional elements were kept the same for all treatments. The EC in the different Mg-f treatments was 0.84–1.78 dS/m, and the pH of the fertigation solution was 5.62–6.95.

### 2.3. Foliar Application of Salt Solutions to Potted Sweet Basil Plants (Experiments B1 [Ca-s, Mg-s, K-s])

Sweet basil was transplanted into pots containing growth mixture, as described above, unless noted otherwise. Fertigation was carried out with the fertilizer 4-2-6 (N-$P_2O_5$-$K_2O$) +3% microelements (Gat Fertilizers, Kiryat Gat, Israel) throughout the experiment. Following two shoot harvests, when plants in the experiments reached 45 to 50 cm in height, the foliar treatment was initiated, and plants were artificially inoculated with *P. belbahrii*. Experiments B were repeated twice with 5 replicates each.

### 2.3.1. Spray Applications of Ca and Mg Salts to Sweet Basil Plants (Experiment B1-[Ca-s, Mg-s, K-s])

Experiments were conducted to test the effects of foliar-applied Ca, Mg and K on SBDM, as generally described for the B1 experiments. The effects of spray solutions containing 1% $CaCl_2$, $MgCl_2$ or KCl (90, 105 and 134 mM Ca, Mg and K, respectively) were each compared with the effects of a spray solution containing 1% NaCl (171 mM Na). Six foliar sprays were applied twice a week for a period of three weeks. At each application, 5 mL of spray solution were applied to each plant using a hand sprayer that emits fine drops, until runoff.

### 2.3.2. Spray Applications of K Salt Solutions to Sweet Basil Plants (Experiments B2-K-s)

To examine the effect of foliar applications of K on SBDM, experiments were conducted under the same conditions as described for the B1 experiments. Four K concentrations (0%, 0.5%, 1.0% and 1.5%, corresponding to 0, 67, 134 and 201 mM K, respectively, as KCl, and 0, 57, 114, and 171 mM K as $K_2SO_4$ (were used in spray solutions in which the salts were KCl or $K_2SO_4$ (Fertilizers & Chemical Compounds Ltd., Haifa, Israel). Six foliar sprays

were applied twice a week for a period of three weeks. At each application, 5 mL of spray solution were applied to each plant using a hand sprayer that emits fine drops, until runoff.

### 2.3.3. Fertigation in Combination with the Foliar Application of K (Experiment B3 [Ca-f, Mg-f, K-s])

Plants potted in perlite were fertigated with a fertilizer containing 4-2-6 (N-$P_2O_5$-$K_2O$) +3% microelements (Gat Fertilizers, Kiryat Gat, Israel) throughout the experiment as described above. Two Ca concentrations in the fertigation solution were evaluated: 1.5 and 3.9 mM Ca. In addition, two Mg concentrations in the fertigation solution were evaluated: 0.3 and 2.0 mM. Spray treatments containing 1% $K_2SO_4$ (114 mM K) were applied to the plants, to runoff, twice a week for three weeks, as described above. The K concentration in the fertigation solution ranged from 1.87 to 2.06 mM. The EC was 1.05–1.16 dS/m and the pH was 6.69–7.47.

### 2.4. Greenhouse Experiments (C)

At Site C, experiments were carried out in a polyethylene-covered greenhouse. Sweet basil plants were planted in perlite (medium size, 1.2 mm, Agrifusia) growth medium in polystyrene containers ($1.0 \times 0.8 \times 0.17$ m), with 24 plants per container. Plants were irrigated daily according to local extension service recommendations. During the initial 5 days, plants were sprinkler-fertigated with 4.3 mM N (10% $NH_4^+$), 1.6 mM K and 0.65 mM P in the fertigation solution to aid their establishment. After that initial period, the plants were irrigated through drippers and fertilized with 8.57 mM N, 3.2 mM K and 0.65 mM P in water until the fertigation treatments were initiated, as described below. Fertigation was performed from 1000-L tanks dedicated to each treatment, with a 17-mm drip-irrigation pipe that had a 2-L/h dripper embedded every 20 cm along its length. Spray treatments were carried out with a backpack sprayer equipped with a conical nozzle. Sprays were administrated until runoff, once or twice a week. Experiments were carried out in the autumn (September–January) or spring (February–June) growing seasons. The experiment was conducted in randomized blocks with four replicates. Each replicate consisted of three containers.

SBDM usually appears 45–60 days after planting, reaches a peak of severity and then begins to become less severe. In the spring growing season, it declined to a low level due to the high temperatures and low humidity typical in the summer (Figure 1a). In the autumn growing season, SBDM severity decreased due to low temperatures (Figure 1b). The results presented from the greenhouse experiments refer to SBDM severity at a certain time after planting or to AUDPC over a certain period.

Chemical fungicide treatments included a rotation of two types of treatments. The first type of treatment was administrated soon after harvest and consisted of a spray application of a mixture of two fungicides: Canon (potassium phosphite, 780 g/L, Luxembourg Industries (Pamol) Ltd., Tel Aviv, Israel applied at 0.3%) + Cabrio Duo (dimethomorph 72 g/L + pyraclostrobin 40 g/L, BASF, Ludwigshafen, Germany applied at 0.05%) (or alternatively, Infinito (flupicolide 62.5 g/L + propamocarb-HCL 625 g/L), Bayer AG, Germany, applied at 0.1%). The second type of treatment was one spray application of Canon at one week before harvest.

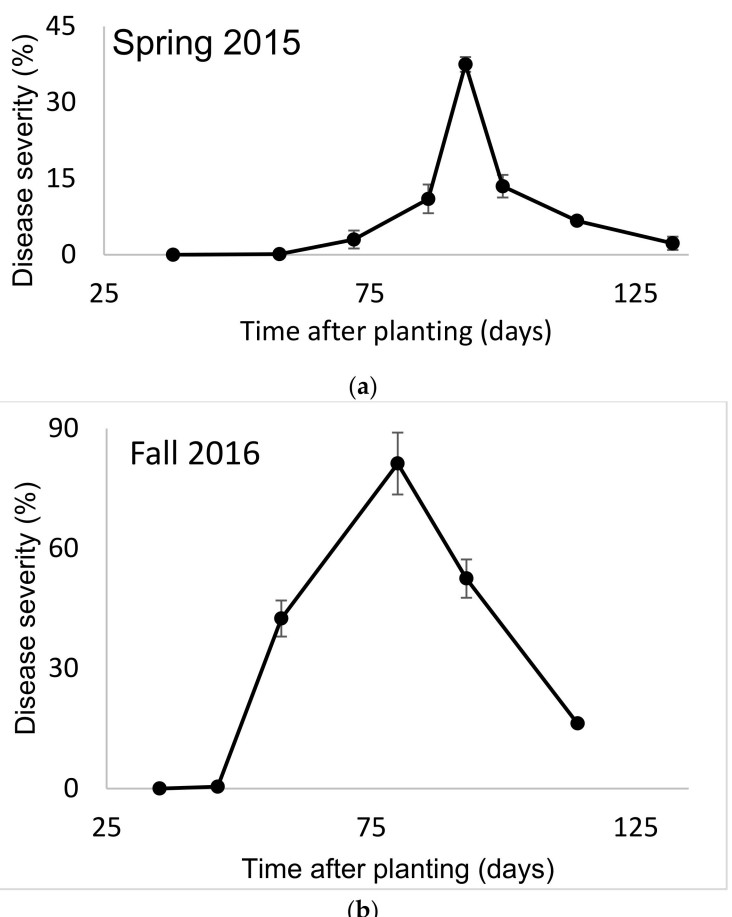

(a)

(b)

**Figure 1.** Development of sweet basil downy mildew (SBDM) during (**a**) a spring growing season and (**b**) an autumn growing season. SBDM severity was evaluated on a scale of 0–100%, in which 0 = healthy plants and 100% = plants completely covered with SBDM symptoms/signs. Bar = SE; *n* = 4.

2.4.1. Effects of a Combination of Ca and Mg in the Fertigation Solution and Foliar-Applied K (Experiment C1a, Spring 2015)

To test the effects of Ca and Mg on SBDM during the spring season, sweet basil was planted on 5 March 2015. Fertigation treatments were started on 29 March 2015. The basic fertilizer described above was the same in all treatments. Irrigation water contained 2.0 mM Ca and 1.6 mM Mg. Addition of Cl salts of Ca and Mg was used for treatments with higher concentrations of these ions in the irrigation water. There were six fertigation treatments to asses all possible combinations of three levels of Ca (2.0, 4.0 and 6.0 mM) and two levels of Mg (1.6 and 4.2 mM). Those treatments were applied with and without a 1% $K_2SO_4$ spray treatment (114 mM K); the plots that received the +/− K-spray treatments were located adjacent to each other. The EC of the fertigation solutions for the six treatments were between 1.81 and 2.70 dS/m, and their pH ranged between 6.85 and 7.06. In 2015, sweet basil shoots were harvested on 31 March, 7 April, 15 April, 22 April, 4 May, 19 May, 2 June, 18 June and 2 July.

2.4.2. Combined Mg, Ca and K Treatments under Commercial Greenhouse Conditions (Experiment C1b, Autumn 2015–2016)

To examine the effects of Ca and Mg on SBDM during the autumn–winter season, sweet basil was planted on 1 September 2015. Fertigation treatments were started on 24 September 2015. The basic fertilizer mentioned above was the same in all treatments. There were six fertigation treatments, to account for all possible combinations of two concentrations of Mg (1.6 and 4.2 mM) and three concentrations of Ca (2.0, 4.0, and 6.0 mM).

Those treatments were applied with and without a 1% $K_2SO_4$ spray treatment (114 mM K); the plots receiving the +/− K spray treatments were located adjacent to each other. The EC of the fertigation solutions in the six treatments ranged from 1.91 to 2.45 dS/m, and their pH values ranged from 6.95 to 6.96. Shoots were harvested on 14 October 2015, 1 November 2015 and 7 January 2016.

### 2.4.3. Effects of Different Concentrations of Mg in the Fertigation Solution under Greenhouse Conditions (Experiment C2a, Autumn 2016)

To test the effect of Mg concentration on SBDM in an autumn crop, sweet basil was planted on 21 September 2016. Fertigation treatments started on 28 September 2016. The basic fertilizer mentioned above was the same in all treatments (Experiment C). The basic Mg concentration was 1.6 mM, and additional treatments of 3.30 and 4.95 mM were made by adding $MgCl_2$ to the irrigation water. Those treatments were applied with and without foliar fungicides, as described above; the plots that received the +/− fungicide treatments were located adjacent to each other. Fungicides were spray-applied in the experimental plots starting from 16 November 2016. The EC of the fertigation solutions for the three Mg treatments ranged from 1.91 to 2.74 dS/m, and their pH values ranged between 6.70 and 7.03. In 2016, shoots were harvested on 26 October, 15 November and 19 December.

### 2.4.4. Effects of Mg Fertigation Treatments under Greenhouse Conditions (Experiment C2b, Spring 2017)

To examine the effects of different Mg concentrations on SBDM in the spring season, sweet basil was planted on 28 February 2017. Fertigation treatments were started on 25 April 2017. The basic fertilizer mentioned above was the same in all treatments (Site C). The basic Mg concentration was 1.6 mM, and additional treatments of 3.3 and 5.0 mM were made by adding $MgCl_2$ to the fertigation solution. Treatments were applied with and without foliar fungicides, as described above; the plots that received the +/− fungicide treatments were located adjacent to each other. Fungicides were sprayed in the experimental plots starting from 30 May 2017. The EC of the fertigation solutions for the three Mg treatments ranged between 1.85 and 2.87 dS/m, and their pH values ranged from 6.95 to 7.12. In 2017, shoots were harvested on 8 May, 29 May and 18 June.

### 2.4.5. Comparison of the Effects of the Different Anions ($Cl^-$ vs. $SO_4^{2-}$) in the Fertigation-Applied Mg Salt (Experiment C3, Autumn 2017–2018)

To examine the effect of the type of anion that is part of the added Mg salt on SBDM, sweet basil was planted on 26 September 2017. The basic fertilizer mentioned above was the same in all treatments (Site C). Starting on 25 October 2017, 1.65 mM $MgCl_2$ and $MgSO_4$ were added to the water, so that the final Mg concentration was 3.3 mM. Treatments were also applied with and without fungicides, as described above; the plots that received +/− fungicide treatments were located adjacent to each other. Fungicides were sprayed in the experimental plots starting from 9 October 2017. The EC of the fertigation solutions for the Mg treatments were between 1.91 and 2.32 dS/m, and their pH values ranged from 6.75 to 6.98. The shoots were harvested on 29 October 2017, 27 December 2017 and 24 January 2018.

### 2.5. Mineral Analysis

In all experiments, shoots were sampled randomly at harvest time from potted plants (Site A) and the commercial-container experiments (Site C) for determination of mineral concentrations. The shoots were rinsed with distilled water and dried in an oven at 70 °C for 48 h. The dried plant material was ground and subjected to chemical analysis. The total K, Mg and Ca concentrations of the shoots were determined after digestion with sulfuric acid and peroxide [30]. K, Na and Mg were analyzed with an atomic absorption spectrophotometer (Perkin-Elmer 460, Norwalk, CT, USA). Ca was analyzed by digestion with nitric acid and perchlorate, followed by atomic absorption spectrophotometry Perkin-Elmer 460,

Norwalk, CT, USA), following [31]. Chloride was extracted from the shoots in water (100:1 water:dry matter) and quantified using a chloride analyzer (model 926, Sherwood).

### 2.6. Infection with P. belbahrii and Evaluation of SBDM Severity

Conidia of *P. belbahrii* were collected in water by washing conidiating leaves of sweet basil plants that were kept in an experimental greenhouse at the Volcani Center. The canopy of potted sweet basil plants was inoculated with a conidial suspension that contained $10^3$ cells mL$^{-1}$ on the afternoon of the inoculation date. The plants were incubated at high RH (>95%) in the dark in a growth chamber at $22 \pm 1$ °C for 12 h, and then incubated in a greenhouse chamber at $22 \pm 2$ °C for 1 week, incubated at high RH (>95%) in the dark in a growth chamber at $22 \pm 1$ °C for 12 h, and then incubated in a greenhouse chamber at $22 \pm 2$ °C for symptom development. Potted sweet basil plants subjected to this artificial inoculation served as inoculum sources to ensure even inoculum loads across the greenhouse in the pot (Experiment A). SBDM severity was evaluated on the sweet basil canopy using a scale of 0–100. Symptoms/signs included chlorosis, dry necrotic lesions and sporulation on the lower leaf side. In this scale, 0 = no signs or symptoms, and 100 = entire surface displays signs and/or symptoms [27].

In the commercial greenhouse experiments (Site C), SBDM epidemics developed naturally during each experimental period. The evaluation of SBDM severity in the greenhouse plots included all plants, except for those along the edges of each plot. SBDM severity was determined every 2 to 3 weeks in each plot of each experiment, on a scale of 0 to 100, in which 0 = all plants visually healthy and 100 = all leaves on all plants in the plot show typical SBDM symptoms of chlorosis or dry necrotic lesions, or signs of sporulation of *P. belbahrii* on their lower side [27].

The germination of *P. belbahrii* conidia was examined on leaves sampled from plants that had been sprayed with 1% KCl, K$_2$SO$_4$, NaCl (134, 114, and 171 mM, respectively) or water with no added salt. Four leaves from each of five plants per treatment were detached, placed on wet paper towel in a Petri dish and inoculated with 20 μL drops containing $10^5$/mL conidia. The inoculated leaves were incubated for 5 h in the dark at 21 °C. Following staining with aniline blue (a mixture of methyl blue and water blue), 50 conidia in each drop were evaluated. The percent germination of the conidia and germ-tube length was evaluated.

### 2.7. Shoot Weight

Batches of shoots that were >15 cm long were harvested and weighed. At Site A, yield was measured for each planted pot. In the commercial greenhouse experiments, yield data was collected separately for each plot and sorted by quality grade. Results are presented in terms of grade A produce per m$^2$.

### 2.8. Data Analysis

The correlations between the concentration of a nutritional element and SBDM severity or between shoot concentrations of two selected nutritional elements were calculated using all individual pairs of data. Linear, exponential, logarithmic and polynomial correlations were examined. The results present the formulas describing these types of correlations, correlation coefficient (*r*) values and $\alpha$ significance levels.

A and B experiments were performed 3 times. Data was combined from replicate experiments since no interactions between trial repetitions were observed. Data in percentages were arcsine-transformed before further analysis. Area under the disease progress curve (AUDPC) values were calculated. Standard errors of the mean (SEs) were calculated and presented alongside the degrees of freedom (*df* = *n* − 1 for controlled-conditions experiments and *df* = *n* − 2 for correlations calculated for greenhouse-conditions data). SBDM severity and AUDPC data were analyzed using ANOVA and Tukey–Kramer's HSD test. The statistical analysis was performed using JMP 10.0 software (SAS Institute, Cary, NC).

Disease reduction was calculated as follows:

$$\% \text{ disease reduction} = 100 - 100 \times (\text{disease severity }_T / \text{ disease severity }_{control}) \quad (1)$$

where T = severity in the treatment and control = severity in the untreated control.

The combined effect of the control measures used was estimated using the Abbott formula [32,33]. The expected reduction in SBDM severity (control efficacy) and the combined suppressive activity were calculated according to the following formula:

$$CE_{exp} = a + b - a \times b/100 \text{ and } SF = CE_{obs}/CE_{exp} \quad (2)$$

where a = severity reduction induced by one measure when applied alone; b = disease reduction induced by the other measure when applied alone; $CE_{exp}$ = expected control efficacy of the combined treatment, if the two measures act additively; $CE_{obs}$ = observed disease reduction for the combined treatment; and SF = the synergy factor achieved by the combined treatment. When SF = 1, the interaction between the control measures is additive; when SF < 1, the interaction is antagonistic and when SF > 1, the interaction is synergistic [32–34]. This same formula was used to calculate SF in the context of yield.

## 3. Results

### *3.1. Pot Experiments—Supplemental Nutrients in the Fertigation Solution*

3.1.1. Effect of K Concentration in the Fertigation Solution on SBDM (Experiment A-K-f)

Raising the concentration of K in the fertigation solution from 0.5 to 5.1 mM increased the concentration of K in the shoots (Figure 2a), as well as a SBDM severity (Figure 2b). Furthermore, the SBDM severity observed at the highest K concentration was significantly higher than that observed at the other concentrations (Figure 2b). Consequently, SBDM was also correlated with the K concentration in the shoots (Figure 2c). The shoot yield was low at the two lower K concentrations, while at 4.11 to 5.79 mM K, there were no significant changes in yield between treatment (Figure 2d).

3.1.2. Effect of the Ca concentration in the Fertigation Solution on SBDM (Experiment A-Ca-f)

Raising the concentration of Ca in the fertigation solution from 0.5 to 6.0 mM increased the Ca concentration in the shoots and decreased SBDM severity (Figure 3a,b). Consequently, SBDM severity was negatively correlated with the Ca concentration in the shoots (Figure 3c). The shoot yield was highest in the 1.56 mM Ca treatment (Figure 3d).

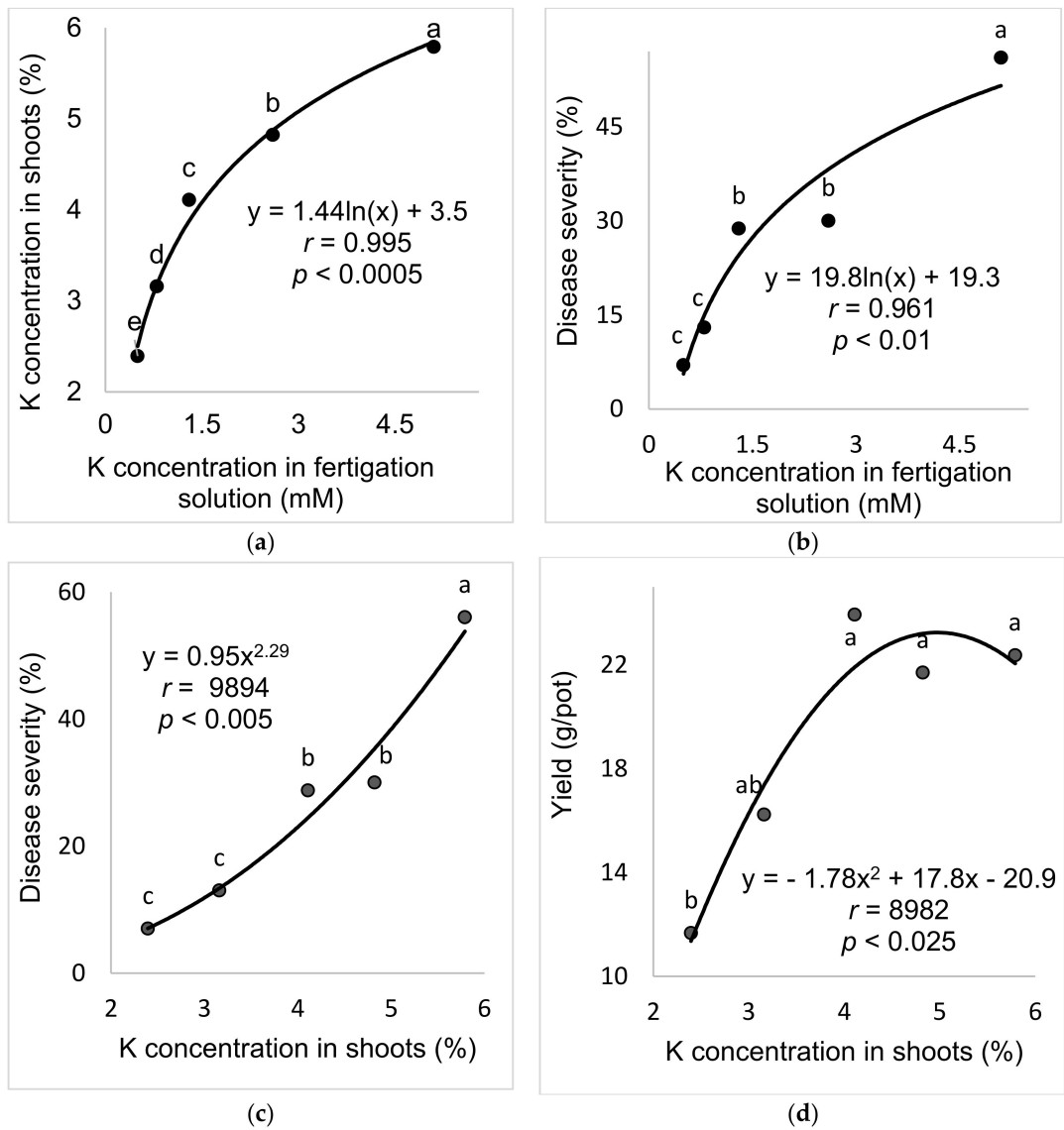

**Figure 2.** Effects of the K concentration in the fertigation solution on (**a**) the K concentration in the sweet basil shoots, (**b**) the severity of sweet basil downy mildew (SBDM) and (**d**) yield. SBDM is described in relation with the K concentration in (**b**) the fertigation solution and (**c**) the shoots. SBDM was evaluated on a 0–100% severity scale, in which 0 = healthy plants and 100% = plants completely covered by SBDM symptoms/signs. Data derive from 3 trials (*n* = 10). In each graph, values for each K treatment followed by a common letter are significantly not different from each other according to Tukey–Kramer's HSD test (*p* ≤ 0.05). The correlation formulas and Pearson regression values (*r*) are presented along with significance levels (*P*). Bar = SE.

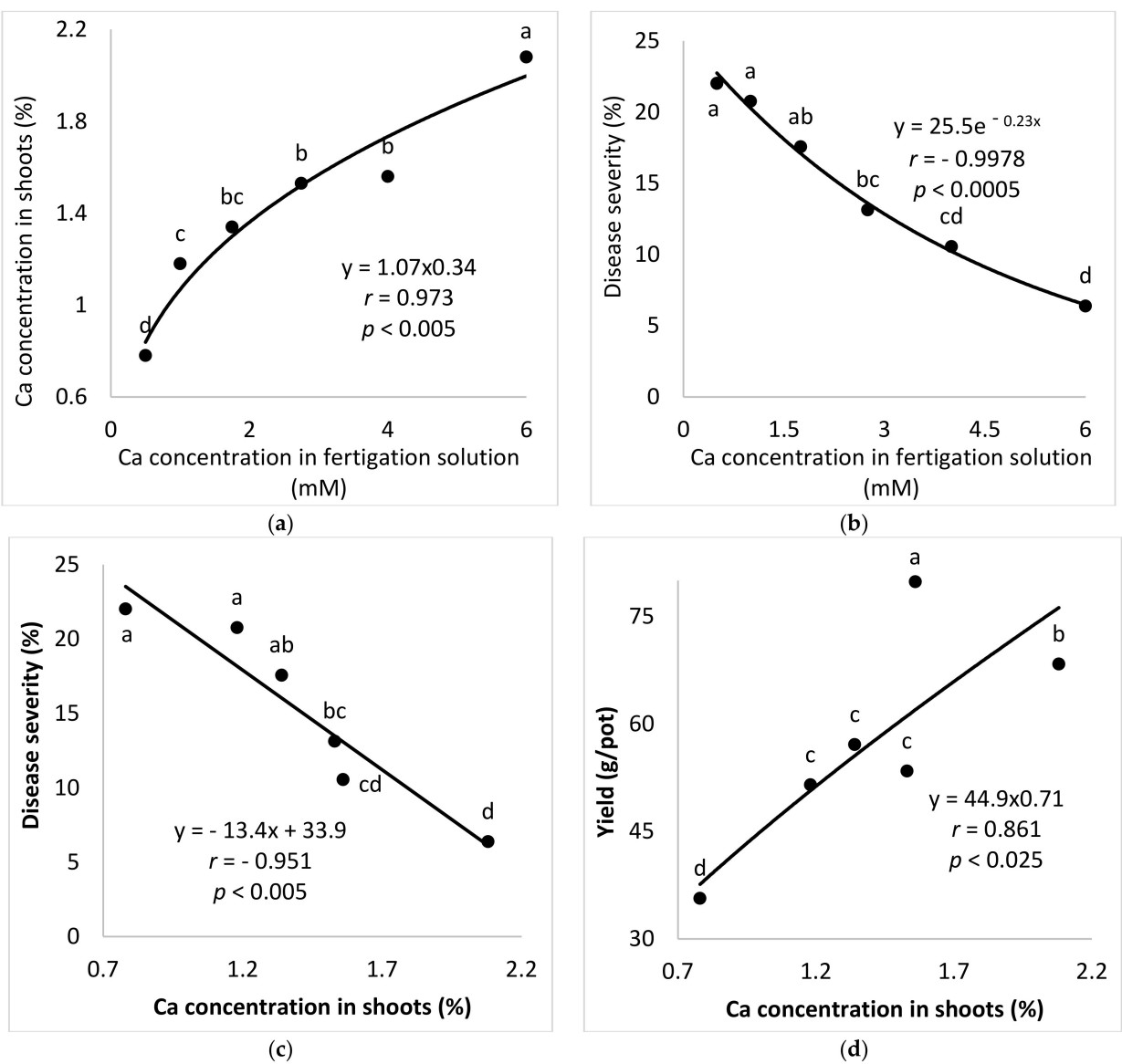

**Figure 3.** Effects of the Ca concentration in the fertigation solution on (**a**) the Ca concentration in the sweet basil shoots, (**b**) the severity of sweet basil downy mildew (SBDM) and (**d**) yield. SBDM severity is described in relation to the Ca concentration in (**b**) the fertigation solution and (**c**) the shoots. SBDM severity was evaluated on a 0–100% scale, in which 0 = healthy plants and 100% = plants completely covered by SBDM symptoms/signs. Data derive from 3 trials (*n* = 10). In each graph, values for each Ca treatment followed by a common letter are significantly not different from each other according to Tukey–Kramer's HSD test (*p* ≤ 0.05). The correlation formulas and Pearson regression values (*r*) are presented along with significance levels (*P*). Bar = SE.

### 3.1.3. Effect of Mg Concentration in the Fertigation Solution on SBDM (Experiment A-Mg-f)

Raising the concentration of Mg in the fertigation solution from 0 to 4.9 mM resulted in similar increases in shoot Mg concentrations for the four highest Mg treatments (Figure 4a). A decrease in SBDM severity was observed in response to higher concentrations of Mg in the fertigation solution and in the shoots (Figure 4b,c). Relatively low shoot yields were observed for the two lower Mg treatments (Figure 4d).

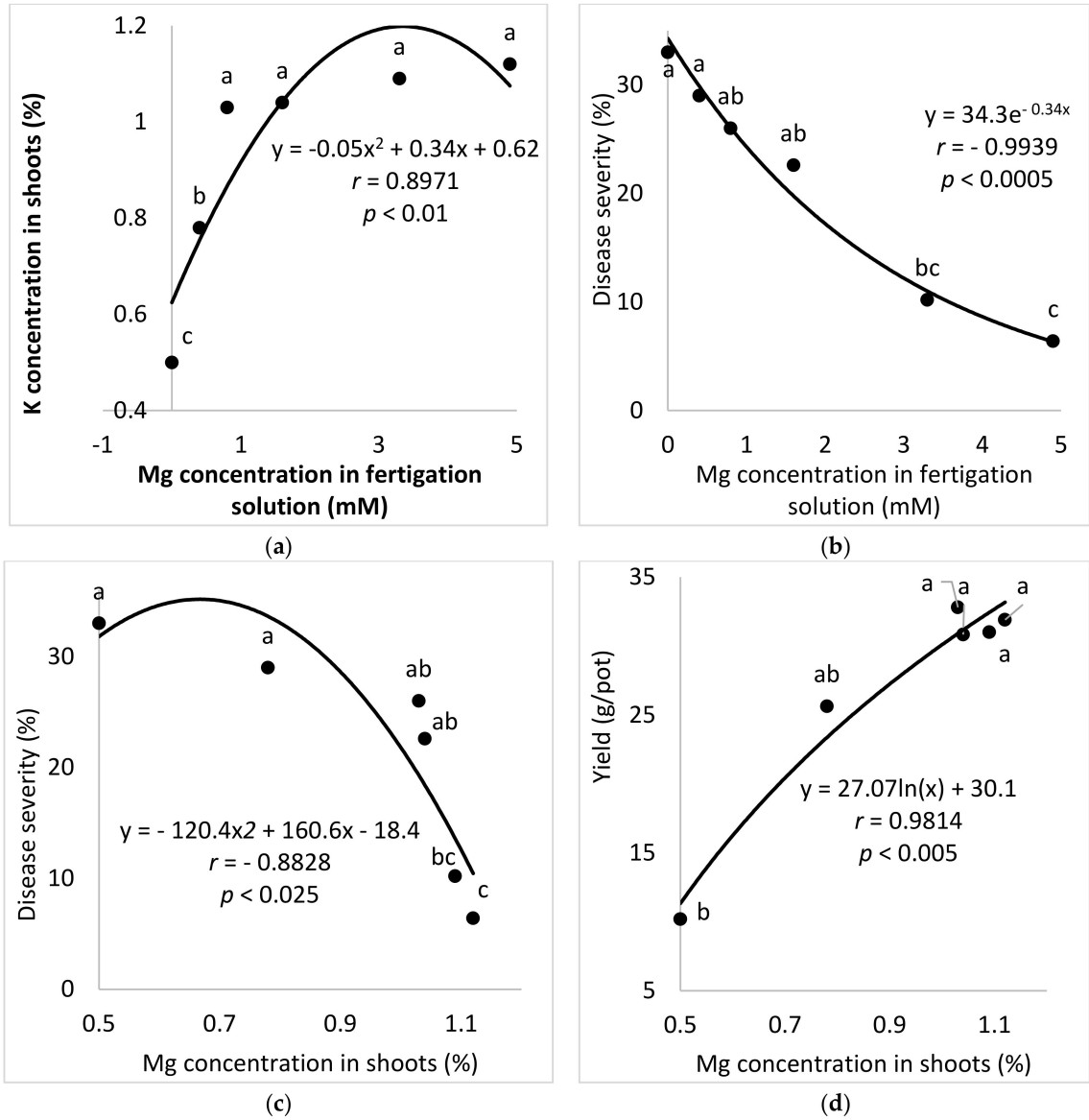

**Figure 4.** Effects of the Mg concentration in the fertigation solution on (**a**) the Mg concentration in the sweet basil shoots, (**b**) the severity of sweet basil downy mildew (SBDM) and (**d**) yield. SBDM severity is described in relation to (**b**) the Mg concentration in the fertigation solution and (**c**) the shoots. SBDM severity was evaluated on a 0–100% scale, in which 0 = healthy plants and 100% = plants completely covered by SBDM symptoms/signs. Data derive from 3 trials ($n = 10$). In each graph, values for each Mg treatment followed by a common letter are significantly not different from each other according to Tukey–Kramer's HSD test ($p \leq 0.05$). The correlation formulas and Pearson regression values (*r*) are presented along with significance levels (*P*). Bar = SE.

### 3.2. Pot Experiments—Foliar Applications of Nutrients (B)

3.2.1. Effects of Foliar-Applied Ca, Mg and K on SBDM Severity (Experiment B1[-Ca-s, Mg-s, K-s])

In separate experiments, sweet basil plants were sprayed with salts of Ca, Mg and K containing 1% Cl$^-$. NaCl served as a control chloride salt. Spray applications of $CaCl_2$, $MgCl_2$ and KCl significantly reduced SBDM severity (71%, 58% and 40%, respectively, Figure 5a–c). NaCl did not reduce SBDM severity in any of our tests (Figure 5).

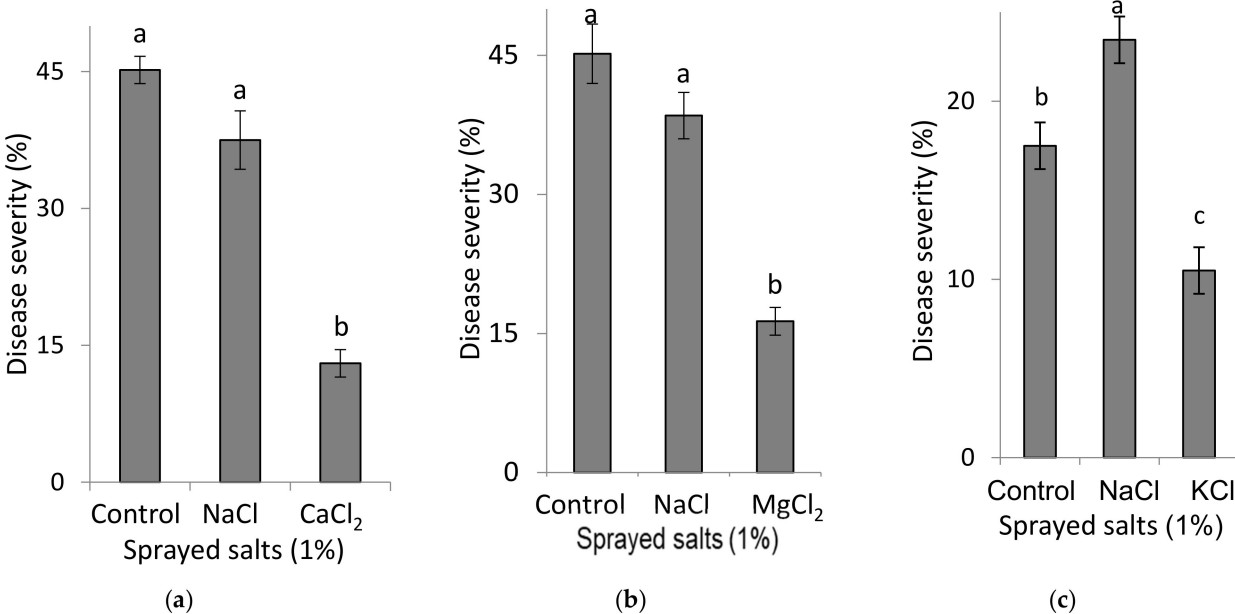

**Figure 5.** Effects of foliar applications of 1% salt solutions on the severity of sweet basil downy mildew (SBDM): (**a**) NaCl and CaCl$_2$, (**b**) NaCl and MgCl$_2$ and (**c**) NaCl and KCl. SBDM severity was evaluated on a 0–100% scale, in which 0 = healthy plants and 100% = plants completely covered by disease symptoms/signs. Data derive from 3 trials (*n* = 5). Values followed by a different letter are significantly different according to two-way ANOVA with Tukey's HSD. Default significance levels were set at *α* = 0.05. Bar = SE.

### 3.2.2. Effect of Foliar-Applied K Salts on SBDM Severity (Experiment B2-K-s)

While higher K concentrations in the fertigation solution and shoots were associated with more severe SBDM, we also wanted to test the effects of different concentrations of foliar-applied K salts on SBDM severity. To that end, KCl and K$_2$SO$_4$ solutions were sprayed on sweet basil plants (Table 2). The interaction between salt type and salt concentration was insignificant. A higher salt concentration resulted in a higher K concentration in the shoots (Table 2). Foliar applications of higher levels of K salts resulted in significantly less severe SBDM (Table 2). The two different anions had no differential effects on SBDM severity or the concentration of K in the shoots (Table 2). The concentrations of N, P, Ca and Mg in the shoots were not affected by these spray treatments (results not presented). The weight of the shoots was not affected by the spray treatments (results not presented). Phytotoxicity symptoms were observed on the leaves that received the 1.5% KCl spray treatment.

**Table 2.** Effects of foliar applications of KCl and K$_2$SO$_4$ on the K concentration in the shoots and the severity of sweet basil downy mildew (SBDM).

| K Salt | Salt Concentration (%) | | | |
|---|---|---|---|---|
| | 0 | 0.5 | 1.0 | 1.5 Avg. (%) |
| K conc. in shoot (%) | | | | |
| KCl | | 5.33 | 6.17 | 7.63   6.37 a |
| K$_2$SO$_4$ | 4.26 | 5.01 | 5.95 | 7.51   6.16 a |
| Avg. conc. (%) | 4.26 d | 5.17 c | 6.06 b | 7.57 a |
| **SBDM severity (%)** | | | | |
| KCl | | 55.0 | 37.4 | 23.0   38.5 a |
| K$_2$SO$_4$ | 87.4 | 41.4 | 40.0 | 26.0   35.3 a |
| Avg. conc. (%) | 87.4 a | 47.5 b | 38.7 c | 24.5 d |

Note: Values followed by a different letter are significantly different according to two-way ANOVA with Tukey's HSD test. Default significance levels were set at *α* = 0.05.

Germination of *P. belbahrii* conidia on leaves of sweet basil was examined following foliar applications of 1% KCl, K$_2$SO$_4$, NaCl and water with no added salt. Conidial

germination rates ranged between $13.3 \pm 4.26\%$ and $20.7 \pm 5.72\%$ ($\pm$SE) for the different treatments and germ-tube length ranged between $161.0 \pm 7.92$ and $176.7 \pm 8.65$ μm ($\pm$SE) for the different treatments. There were no significant differences between the treatments for either of those measures of conidial germination ($p \leq 0.05$).

SBDM severity was evaluated on a 0–100% severity scale, in which 0 = healthy plants and 100% = plants completely covered by disease symptoms/signs. Data derive from 3 trials ($n$ = 5). The interaction of the major parameters (K salt concentration × anion) did not significantly affect the K concentration in the shoots or SBDM severity. Therefore, the values for the major treatments are presented and analyzed. Values in each parameter followed by a different letter are significantly different according to two-way ANOVA with Tukey's HSD. Default significance levels were set at $\alpha$ = 0.05.

### 3.2.3. Effects of Fertigation-Applied Ca and Mg in Combination with Foliar-Applied $K_2SO_4$ (Experiment B3 [Ca-f, Mg-f, K-s])

Ca and Mg were added to the fertigation solution, so their concentrations in that solution were 3.9 and 2.0 mM, respectively (Table 3). A $K_2SO_4$ spray was applied in combination with the fertigation treatments. The addition of Ca to the fertigation solution significantly increased the concentration of $Ca^{2+}$ in the shoots and was associated with a slight decrease in the concentration of Mg in the shoots (Table 3). The addition of Mg to the fertigation solution increased the concentration of Mg in the shoots and decreased the concentration of Ca in the shoots (Table 3). The K spray significantly increased the amount of $K^+$ in the shoots in both the Ca and the Mg fertigation treatments (Table 3).

**Table 3.** Effects of the addition of $CaCl_2$ or $MgCl_2$ to the fertigation solution +1% $K_2SO_4$ spray on nutrient concentrations (% dry weight) in sweet basil shoots.

| Nutrient. | Ca Fertigation Treatment (mM), with or without $K_2SO_4$ Spray (K-s) | | | | Mg Fertigation Treatment (mM) with or without $K_2SO_4$ Spray (K-s) | | | |
|---|---|---|---|---|---|---|---|---|
| | Ca, 1.5 | 1.5 | 3.9 | 3.9 | Mg, 0.3 | 0.3 | 2.0 | 2.0 |
| | K-s, − | + | − | + | K-s, − | + | − | + |
| Ca | 1.41 b | 1.33 b | 1.73 a | 1.71 a | 1.42 a | 1.33 a | 1.07 b | 0.91 b |
| Mg | 0.30 a | 0.28 ab | 0.24 b | 0.26 ab | 0.23 c | 0.28 c | 1.09 a | 0.93 b |
| K | 3.72 b | 5.48 a | 3.72 b | 5.68 a | 3.73 b | 5.48 a | 4.86 b | 5.26 a |

Values in each row followed by a different letter are significantly different according to two-way ANOVA with Tukey's HSD. Default significance levels were set at $\alpha$ = 0.05.

Applications of higher levels of Ca (3.9 mM) and Mg (2.0 mM) reduced SBDM severity (by 38.6% and 51.4%, respectively; Figure 6a,b). The foliar-applied K was associated with a 72.5% reduction in SBDM severity. However the combination of foliar-applied K and increased Ca or Mg in the fertigation solution did not provide disease control that was any better than that provided by each treatment alone (Figure 6a,b). An increased Ca concentration in the fertigation solution did not increase yield (Figure 6c); whereas an increased concentration of Mg in the fertigation solution did increase yield (Figure 6d). The foliar application of K increased yield in the two Ca treatments and in the treatment with the lower Mg concentration (Figure 6).

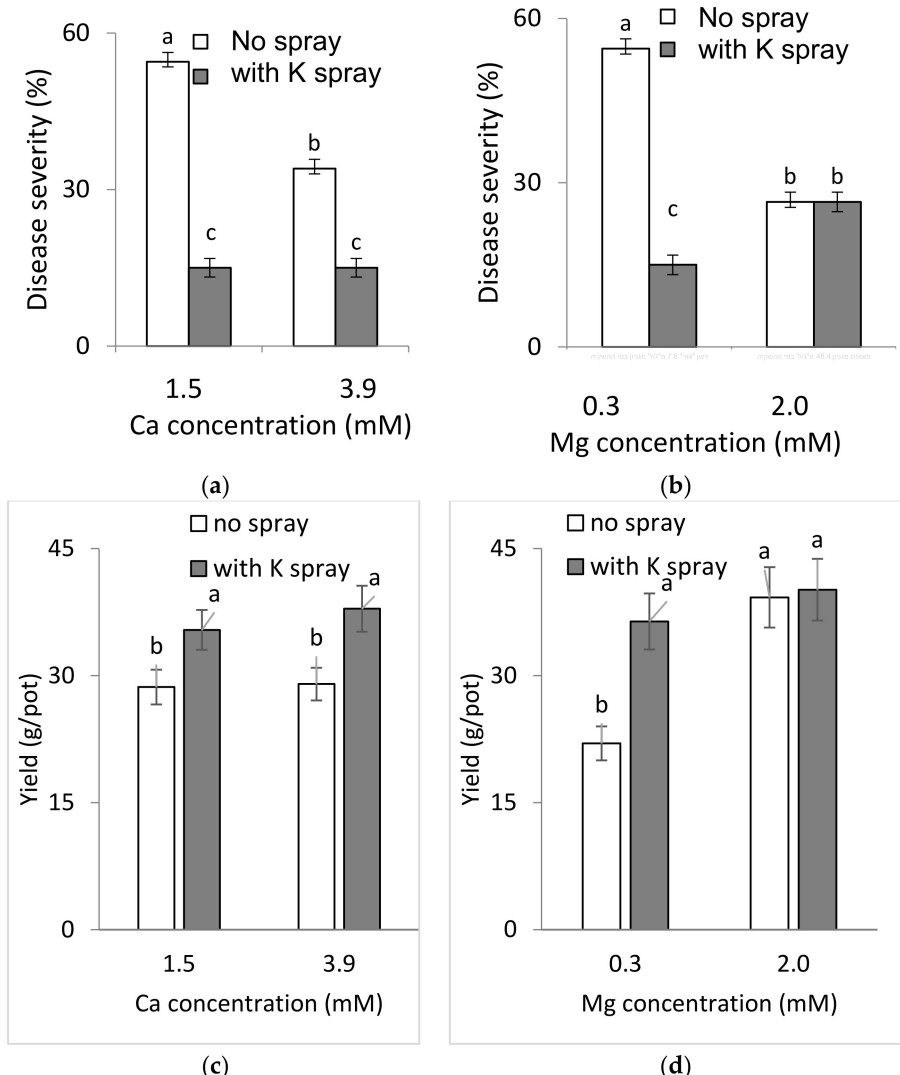

**Figure 6.** Effects of additional Ca (**a**) and (**c**) and Mg (**b**) and (**d**) without (□) and with (■) a foliar application of 1% $K_2SO_4$ on the severity of sweet basil downy mildew (SBDM; a and b) and shoot yield (**c**,**d**). SBDM severity was evaluated on a 0–100% scale, in which 0 = healthy plants and 100% = plants completely covered by SBDM symptoms/signs. Data derive from 3 trials (*n* = 5). Values followed by a different letter are significantly different according to two-way ANOVA with Tukey's HSD test. Default significance levels were set at $\alpha$ = 0.05. Bar = SE.

### 3.3. Commercial Greenhouse Experiments (Site C)

3.3.1. Combined Mg, Ca and K Treatments under Greenhouse Conditions (Experiment C1a, Spring 2015)

The effects of Ca and Mg (as Cl salts) applied through fertigation with and without foliar-applied $K_2SO_4$ were tested under commercial-greenhouse conditions. The Cl concentration in the fertigation treatments was increased as determined by the Ca + Mg salt concentration (Table 4). The concentrations of the elements in the basil shoots were determined in plots that were not sprayed with K solution. The concentration of Cl in the basil shoots was positively correlated with the concentration of Cl in the fertigation solution. This indicates that the plants responded to the treatments as expected. However, the Ca concentration in the shoots accounted for between 3.64% and 4.06% of the shoots dry weight and was similar across the various treatments. An increase in the concentration of Mg in the fertigation solution resulted in an increase in the Mg concentration in the shoots, whereas an increase in the Ca concentration in the fertigation solution was associated with a decrease in the concentration of Mg in the plant tissue. The concentration of K in the

shoots was not significantly affected by the treatments (Table 4). N concentrations in the shoots ranged between 2.91 and 3.28%, and P concentrations in the shoots ranged between 0.33% and 0.39%.

The Ca and Mg fertigation treatments and the foliar application of K affected SBDM severity. ANOVA revealed a significant interaction between the Ca and Mg fertigation treatments. Therefore, the Ca and Mg fertigation treatments were evaluated individually. The higher Ca concentration in the fertigation solution significantly suppressed SBDM (67.2–82.7%) when no Mg was added to the fertigation solution (Figure 7a). The higher-Mg treatment with no extra Ca provided 63.9% suppression (Figure 7a). The combination of supplemental Ca and Mg did not provide any more SBDM suppression than either of those treatments alone. Foliar application of $K_2SO_4$ suppressed SBDM relative to the unsprayed control (Figure 7b). The combination of foliar-applied K and 4.0 mM Ca in the fertigation solution had a synergetic effect, with a synergism factor (SF) of 1.12. The total yield of grade A harvested shoots ranged between 1.185 and 1.327 $Kg/m^2$ for the various treatments, with no significant differences ($P \leq 0.05$) between treatments (results not presented).

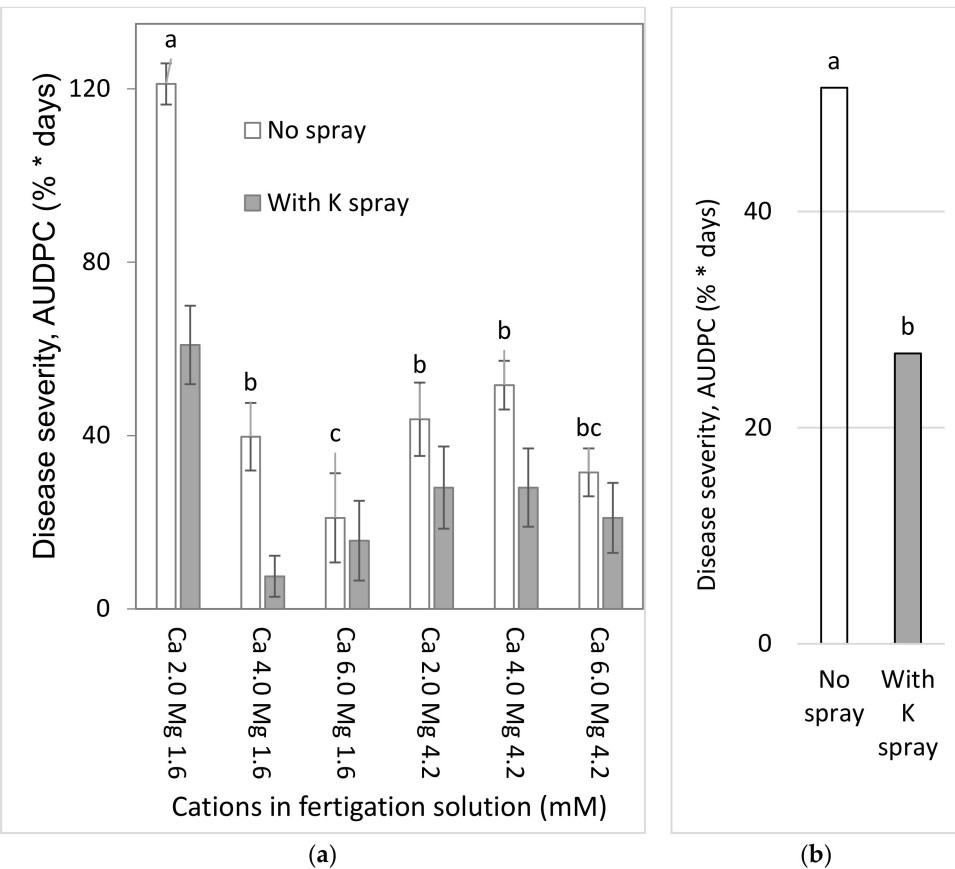

**Figure 7.** Effects of fertigation with three Ca concentrations (2.0–6.0 mM) applied in combination with two concentrations of Mg (1.6–4.2 mM), without (□) and with (■) a foliar spray of 1% $K_2SO_4$, on the severity of sweet basil downy mildew (SBDM). SBDM severity was evaluated on a 0–100% scale, in which 0 = healthy plants and 100% = plants completely covered by SBDM symptoms/signs. The area under disease progress curve (AUDPC) over 86 days was calculated. ANOVA revealed a significant interaction between the Ca and Mg fertigation treatments, and it revealed no interaction between the treatments of no spray and K spray. Therefore, Ca and Mg fertigation treatments are described as single treatment (**a**). The foliar K treatments were analyzed as major treatments over all of the irrigation treatments (**b**). Values in each graph that are labelled with a common letter are significantly not different according to ANOVA with Tukey's HSD. Default significance levels were set at $\alpha$ = 0.05. Bar = SE.

**Table 4.** Concentrations of nutrients in shoots of sweet basil plants treated with Ca and Mg in the fertigation solution with no foliar-applied K under commercial-greenhouse conditions (Experiment C1, Spring 2015 and Experiment C1b, Autumn 2015–2016).

| Ca | Mg | Cl | Ca | Mg | K | Cl | Ca | Mg | K | Cl |
|---|---|---|---|---|---|---|---|---|---|---|
| (mM in Fertigation Solution) | | | (% Dry Weight) | | | | | | | |
| | | | **Experiment C1a** | | | | **Experiment C1b** | | | |
| 2.0 | 1.6 | 6.4 | 4.06 ± 0.401 a [1] | 0.54 ± 0.051 | 5.73 ± 0.687 a | 0.97 ± 0.082 | 3.70 ± 0.361 a | 0.52 ± 0.048 | 4.90 ± 0.481 a | 0.66 ± 0.054 |
| 4.0 | 1.6 | 11.2 | 3.84 ± 0.372 a | 0.45 ± 0.038 | 5.42 ± 0.656 a | 1.33 ± 0.097 | 3.43 ± 0.352 a | 0.51 ± 0.049 | 3.52 ± 0.442 bc | 1.32 ± 0.094 |
| 6.0 | 1.6 | 15.2 | 3.70 ± 0.345 a | 0.38 ± 0.035 | 5.05 ± 0.662 a | 1.65 ± 0.139 | 3.47 ± 0.341 a | 0.42 ± 0.039 | 3.34 ± 0.332 c | 1.45 ± 0.131 |
| 2.0 | 4.2 | 12.4 | 3.65 ± 0.485 a | 0.70 ± 0.067 | 5.81 ± 0.712 a | 1.32 ± 0.125 | 3.43 ± 0.369 a | 0.72 ± 0.065 | 3.64 ± 0.321 bc | 1.32 ± 0.093 |
| 4.0 | 4.2 | 16.4 | 3.68 ± 0.471 a | 0.60 ± 0.051 | 5.37 ± 0.705 a | 1.64 ± 0.151 | 3.53 ± 0.365 a | 0.65 ± 0.054 | 4.61 ± 0.364 a | 1.45 ± 0.137 |
| 6.0 | 4.2 | 20.4 | 3.82 ± 0.481 a | 0.55 ± 0.049 | 5.70 ± 0.743 a | 1.96 ± 0.168 | 3.47 ± 0.357 a | 0.65 ± 0.062 | 3.76 ± 0.313 b | 1.83 ± 0.163 |
| | Ca × Mg interaction: | | yes | no | yes | no | yes | no | yes | no |
| **Major-parameters analysis** | | | | | | | | | | |
| Ca | | | | | | | | | | |
| 2.0 | | | 0.62 a [1] | | 1.15 c | | | 0.62 a | | 0.99 a |
| 4.0 | | | 0.52 ab | | 1.48 b | | | 0.58 ab | | 1.38 b |
| 6.0 | | | 0.46 b | | 1.81 a | | | 0.53 b | | 1.64 c |
| | Mg | | | | | | | | | |
| | 1.6 | | 0.46 b | | 1.32 b | | | 0.48 a | | 1.14 b |
| | 4.2 | | 0.62 a | | 1.64 a | | | 0.67 b | | 1.53 a |

The analyses of elements were conducted on 8 June 2015 (Experiment 1a) and 3 December 2015 (Experiment 1b). Values for single treatments are averages of 4 plot replicates (±SE.). [1] Values followed by a different letter are significantly different according to two-way ANOVA with Tukey's HSD test. Default significance levels were set at α = 0.05. "n" = no-interaction was insignificant for the elements concentration in the [Ca concentration] × [Mg concentration] treatments, thus the values of the major treatments are presented and analyzed.

3.3.2. Combined Mg, Ca and K Treatments under Commercial Greenhouse Conditions (Experiment C1b, Autumn 2015)

In the autumn season, we tested the combination of supplemental Ca and Mg in the fertigation solution and compared their effect to that of a $K_2SO_4$ spray and fungicide sprays that were applied to plants irrigated with water (2.0 mM Ca and 1.6 mM Mg). The concentration of Ca in the fertigation solution was not associated with any increase in the concentration of Ca in the shoots (Table 4). Increasing the Mg concentration in the fertigation solution to 4.2 mM increased the Mg concentration in sweet basil shoots. The combination of additional Mg and Ca in the fertigation solution decreased the Mg concentration in the shoots (Table 4). The concentration of K in the shoots was decreased when the Ca concentration in the fertigation solution was increased (Table 4). Similar to the spring experiments, the Cl concentration in the basil shoots increased according to the fertigation treatments (Table 4) but was not affected by the K treatments or fungicide applications (results not presented). The concentration of N in the shoots ranged between 3.92% and 4.33% and the concentration of P in the shoots ranged between 0.83% and 0.91%.

SBDM severity was significantly suppressed by fertigation with the higher Ca concentration (6 mM) together with the basic Mg concentration (1.6 mM; Figure 8). Unlike the spring experiment, in this experiment, SBDM severity was reduced by the higher Mg concentration (4.2 mM) applied in combination with the two lower Ca concentrations (2 and 4 mM; Figure 8). SBDM severity was also reduced by the foliar K and the fungicide treatments that were applied to plants grown with the basic fertigation; the latter treatment provided superior SBDM suppression (Figure 8).

We examined the relationships between the concentrations of nutritional elements in sweet basil shoots from all single plots (replicates) and SBDM severity. We plotted the shoot concentrations of Ca vs. SBDM at 92 days after planting and the shoot concentrations of Mg vs. SBDM AUDPC 74 to 92 days. The best-fit regression formulas are presented. Pearson regression values ($r$) are presented along with significance levels ($P$). SBDM severity was negatively correlated with the shoot concentrations of Mg (line formula: $y = -1113x^2 + 834x + 190$; $n = 36$; $r = -0.3575$; $p < 0.05$) and Ca (line formula: $y = 323e^{-0.19x}$; $n = 36$; $r = -0.3309$; $p < 0.05$). K concentrations in the shoots were not correlated with SBDM severity.

None of the fertigation or spray treatments had any significant effect on yield (results not presented).

3.3.3. Effects of Mg Treatments under Commercial Greenhouse Conditions (Experiment C2a, Autumn 2016)

Since the interaction between the Ca and Mg fertigation treatments provided improved disease control, in the next two greenhouse experiments, Mg was tested with Ca only at a basic level (2 mM). $MgCl_2$ was added to the fertigation solution for the autumn crop, to reach three concentrations of Mg: 1.65, 3.30 and 5.00 mM. Fungicide sprays were applied in all of those Mg treatments. An analysis of elements in the shoots performed on 16 November 2016 revealed Mg shoot concentrations of 0.98%, 1.02% and 1.29%, respectively; the higher Mg concentration was significantly different from the two lower concentrations ($p < 0.05$). The N concentrations in the shoots were between 5.54% and 5.67%, the P concentrations in the shoots were between 0.86% and 0.96% and the K concentrations in the shoots were between 0.33% and 4.70%, with no significant differences between the three treatments. Cl concentrations in the shoots increased as the Mg level increased, reaching 0.89%, 1.13% and 1.36%, respectively; each treatment was significantly different from the others ($p < 0.05$).

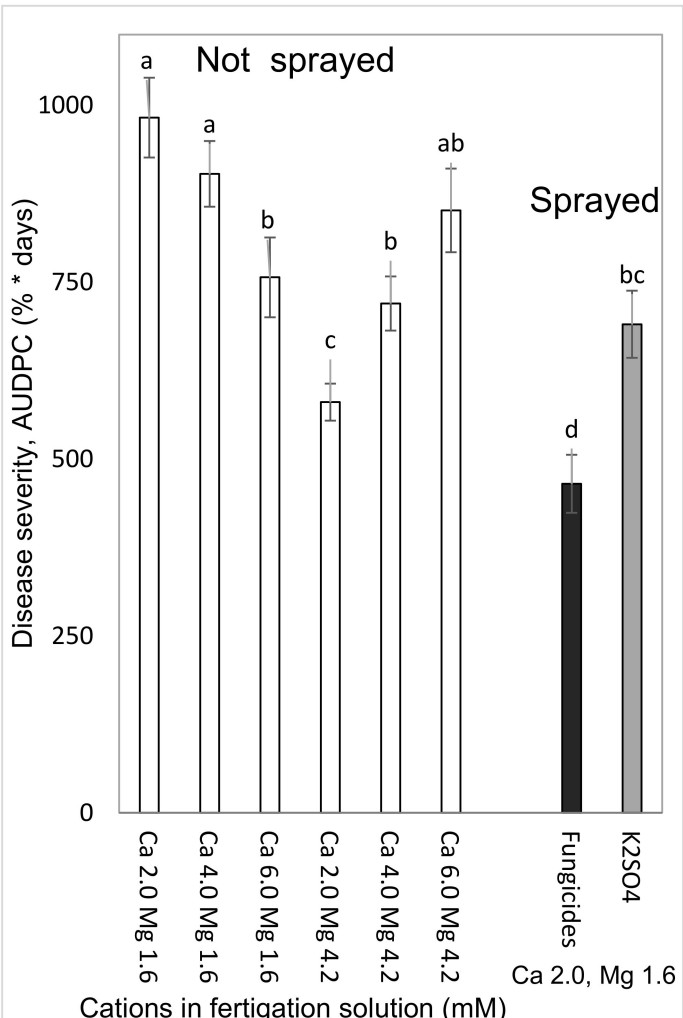

**Figure 8.** Effects of the combinations of three Ca concentrations and two Mg concentrations in the fertigation solution and foliar applications of 1% $K_2SO_4$ and chemical fungicides (in 2 mM Ca and 1.6 mM Mg fertigation treatments) on the severity of sweet basil downy mildew (SBDM). SBDM severity was evaluated on a 0–100% scale, in which 0 = healthy plants and 100% = plants completely covered by SBDM symptoms/signs. The area under disease progress curve (AUDPC) over 42 days was calculated. ANOVA revealed a significant interaction between the Ca and Mg fertigation treatments. Therefore, we present an analysis of the individual treatments. Values followed by a different letter are significantly different according to two-way ANOVA with Tukey's HSD test. Default significance levels were set at $\alpha = 0.05$. Bar = SE.

SBDM severity was significantly reduced (28% to 38%) by the 3.3 and 5.0 mM Mg treatments (Figure 9a, left). The fungicide treatment dramatically reduced SBDM severity, and increased Mg fertigation did not result in any further significant reduction (Figure 9a, right). Calculation of the SF of the reduction in disease severity provided by each of the increased-Mg treatments and the fungicide treatment revealed no synergism between the 3.3 mM Mg treatment and the fungicide treatment but did reveal significant synergism between the 5.0 mM Mg treatment and the fungicide treatment (SF = 1.02).

Shoot yield was 2.406 Kg/$m^2$ in the untreated control. Shoot yield was significantly increased by the fungicides and by the 5.0 mM Mg treatment. In terms of yield, 3.3 mM Mg combined with the fungicide treatment was superior to either of those treatments alone (Figure 9b–d).

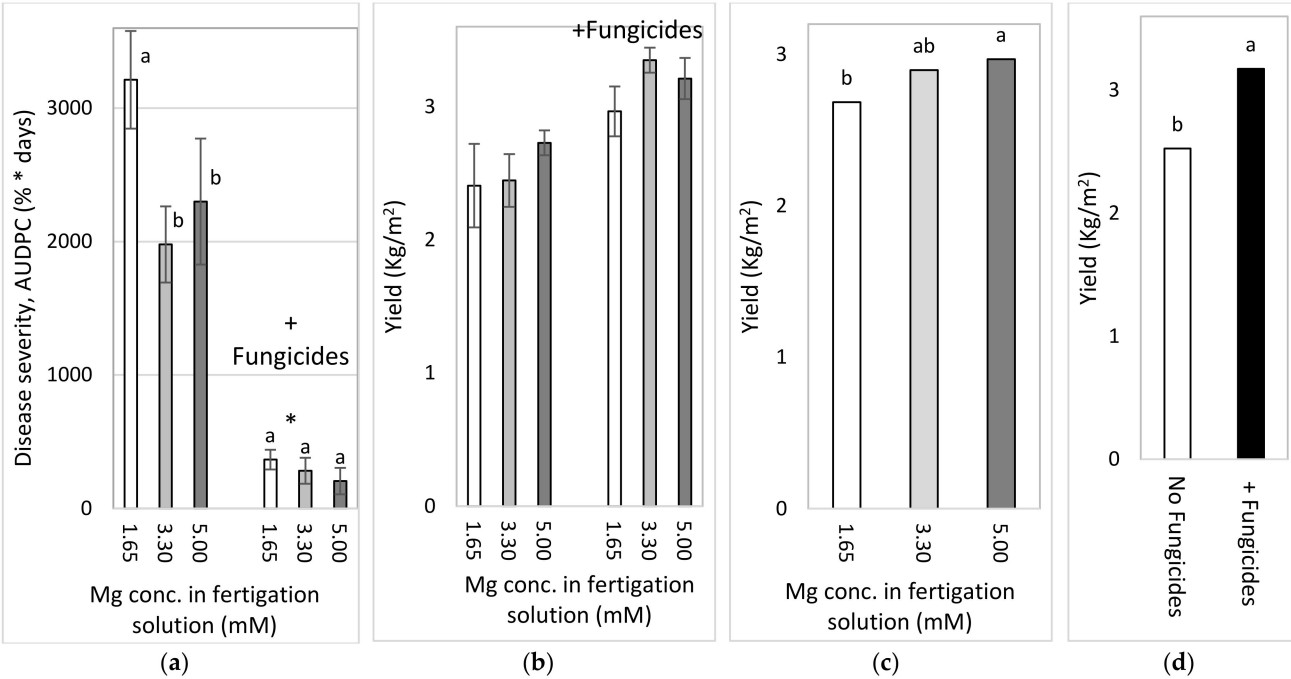

**Figure 9.** Effects of Mg concentrations on (**a**) the severity of sweet basil downy mildew (SBDM) and (**b**–**d**) cumulative shoot weight at 98 days after planting. All of the Mg-treatment plots were either not treated or treated with chemical fungicides (+fungicides). SBDM severity was evaluated on a 0–100% scale, in which 0 = healthy plants and 100% = plants completely covered by SBDM symptoms/signs. The area under disease progress curve (AUDPC) over 114 days was calculated. The interaction between the Mg treatments and the fungicide treatment was significant, so an analysis of SBDM severity is presented for individual treatments. Values for three Mg concentrations in both fungicide treatments followed by a different letter are different from each other (**a**). The interactive effect of the Mg treatments and the fungicide treatment on yield was not significant (**b**), so each major parameter was analyzed independently. Mg (**c**) and fungicide treatments (**d**) followed by a different letter are significantly different according to two-way ANOVA with Tukey's HSD. Default significance levels were set at $\alpha = 0.05$. * SBDM severity in the fungicide treatments was significantly different from the parallel no-fungicide treatments (**a**). Bar = SE.

### 3.3.4. Effects of Mg treatments under Commercial Greenhouse Conditions (Experiment C2b, Spring 2017)

Similar to the autumn experiment, $MgCl_2$ was added to the fertigation solution for a spring crop. Three concentrations of Mg were included in this experiment: 1.65, 3.30 and 5.00 mM. An analysis of elements in the shoots was performed 19 June 2017. The concentrations of Mg in the sweet basil shoots were 0.56%, 0.63% and 0.77%, respectively; the higher Mg concentration was significantly different from the two lower concentrations ($p < 0.05$). The concentration of N in the shoots was between 4.73% and 5.00%, the shoot concentrations of P were between 0.84% and 0.91%, the shoot concentrations of K were between 5.45% and 5.65%, and the shoot concentrations of Ca were between 3.05% and 3.12%, with no significant differences between the three treatments. Cl concentrations in the shoots increased with the 1.65, 3.30 and 5.00 mM Mg treatments to 1.75%, 1.91% and 2.16%, respectively; each treatment was significantly different from the others ($p < 0.05$).

The Mg treatments and the fungicide treatment had no statistically significant interactive effects on SBDM severity (Figure 10a). SBDM was significantly and similarly reduced by the 3.3 mM and 5.0 mM Mg treatments (39.9% and 53.0% reduction, respectively, Figure 10b) and by the fungicide treatment (20.4% reduction; Figure 10c). Calculation of the SF of the reduction in disease severity induced by each of the increased-Mg treatments and the fungicide treatment revealed no synergism between the 5.0 mM Mg treatment and the fungicide treatment. However, there was significant synergism between the 3.3 mM Mg treatment and the fungicide treatment (SF = 1.09).

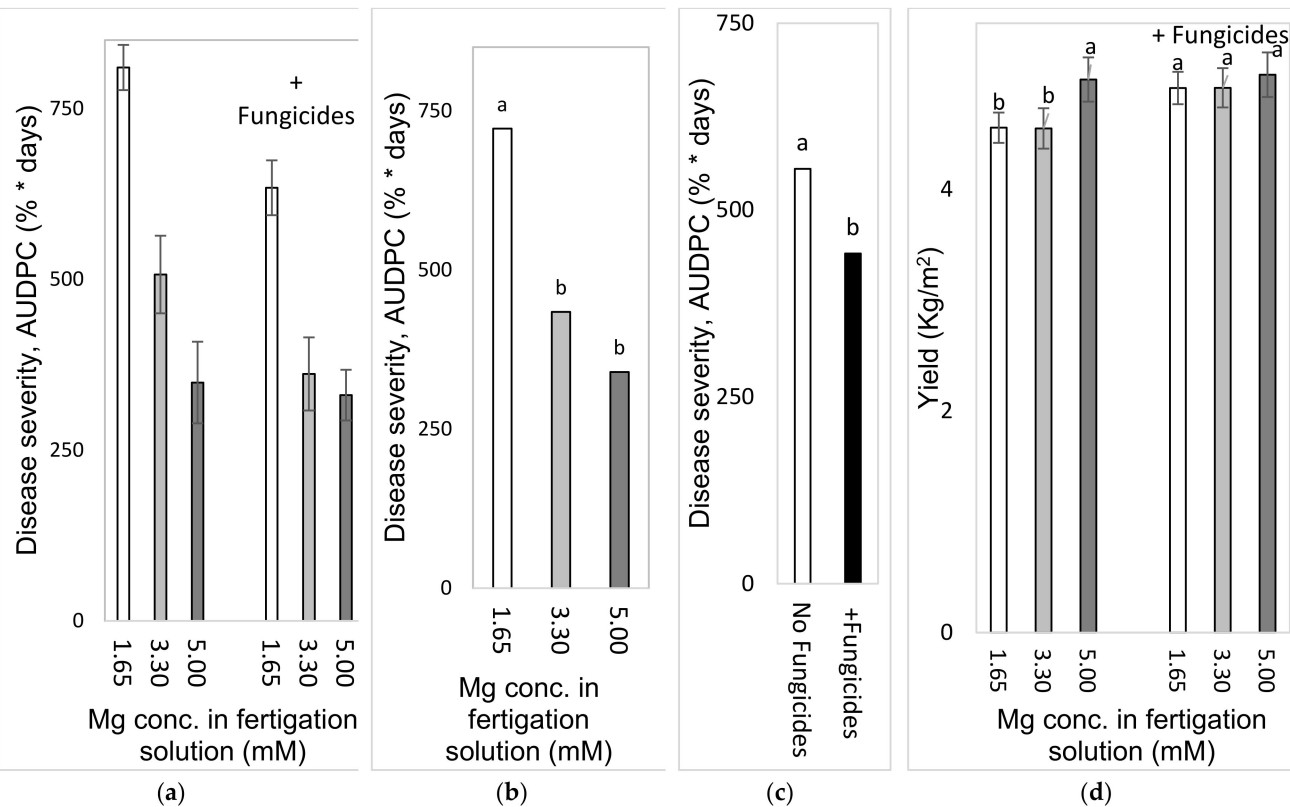

**Figure 10.** Effects of different Mg concentrations on (**a–c**) the severity of sweet basil downy mildew (SBDM) and (**d**) cumulative shoot weight at 110 days after planting. Plots were not treated or treated with chemical fungicides (**a,c,d**). SBDM severity was evaluated on a 0–100% scale, in which 0 = healthy plants and 100% = plants completely covered by SBDM symptoms/signs. The area under disease progress curve (AUDPC) over 63 days was calculated. The interactive effect of Mg treatments and fungicide treatments on SBDM severity was not significant (**a**). Each Mg concentration (**b**) and fungicide treatment (**c**) followed by a different letter is significantly different from the others, according to two-way ANOVA with Tukey's HSD test. The interaction between the Mg treatments and the fungicide treatment significantly affected yield, so an analysis of shoot weight is presented for the individual treatments. Yield values corresponding to the three Mg treatments in both fungicide treatments that are followed by a different letter are significantly different from each other (**d**) according to two-way ANOVA with Tukey's HSD. Default significance levels were set at $\alpha$ = 0.05. Bar = SE.

The Mg treatments and the fungicide treatment did have a statistically significant interactive effect on yield (Figure 10d). Shoot yield was significantly increased by the higher Mg treatment (5.0 mM Mg) when no fungicides were sprayed on the plants. The application of fungicide increased yield similarly to the higher Mg treatment (Figure 10d). The combination of the increased Mg concentration with foliar-applied fungicide did not provide better yields than any of the those treatments alone (Figure 10c,d).

3.3.5. The Effect of the Anion ($Cl^-$ vs. $SO_4^{-2}$) in the Mg Salt Applied via Fertigation (Experiment C3, Autumn 2017)

To test the effect of the anion in the Mg salts that were included in the fertigation solution, we used $MgCl_2$ and $MgSO_4$. The control included 1.65 mM Mg and the increased-Mg treatment had an Mg concentration of 3.3 mM. The analysis of elements in the shoots that was performed on 27 December 2017 revealed Mg concentrations in the sweet basil shoots of 0.52% for the plants that received the basic irrigation solution, an Mg concentration of 0.66% in the shoots of plants treated with $MgCl_2$ and an Mg concentration of 0.69% in the shoots of plants treated with $MgSO_4$. The results for the two salt treatments were significantly different from those for the basic irrigation treatment, with no significant difference between the results for the two salts ($p > 0.05$). The N, P, K and Ca concentrations in the shoots were 4.13–4.44%, 0.71–0.77%, 4.57–4.93% and 2.20–2.36%, respectively, with

no significant differences between the three Mg treatments. The Cl concentration in the shoots was 0.65% for the $MgCl_2$ treatment, which was significantly ($p < 0.05$) higher than the Mg levels of 0.83% and 0.67% that were observed for the basic fertilization treatment and the $MgSO_4$ treatment, respectively.

The Mg treatments and the fungicide treatment had no significant interactive effect on SBDM severity (Figure 11a). SBDM severity was significantly reduced by $MgCl_2$ and $MgSO_4$ applied at a concentration of 3.3 mM (reductions of 44.0% and 23.0%, respectively); the chloride salt provided superior disease control (Figure 11 b). The fungicide treatment reduced SBDM severity by 21.7% (Figure 11c). Calculation of the SF of the disease control provided by each of the fertigation-applied Mg salts and the fungicide treatment revealed no synergism between the supplemental $MgCl_2$ and the fungicide treatment. However, there was significant synergism between $MgSO_4$ and the fungicide treatment (SF = 1.032). We also examined the relationship between nutrient concentrations in the sweet basil shoots and disease severity. That examination revealed a negative correlation between the shoot concentration of Mg and disease severity (Figure 11d).

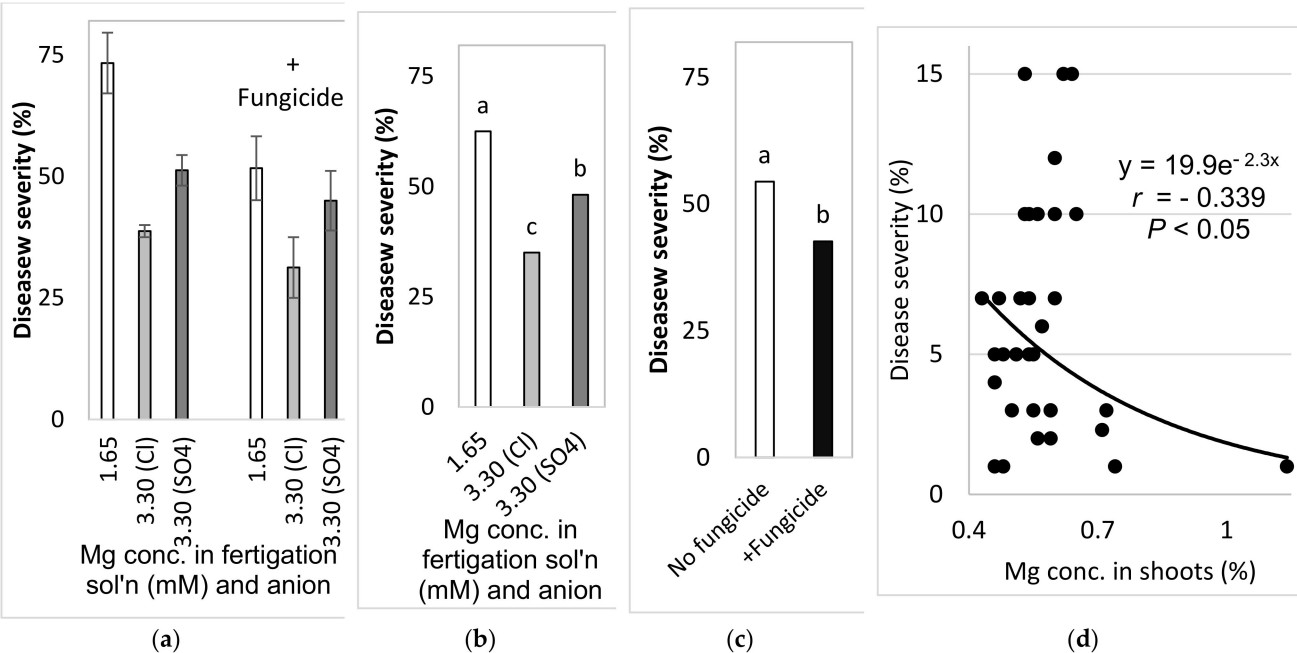

**Figure 11.** Effects of Mg salts ($Cl^-$ vs. $SO_4^{2-}$) on the severity of sweet basil downy mildew (SBDM). Plots were not treated or treated with chemical fungicides (**a,c**). SBDM severity was evaluated at 80 days after planting, using a 0–100% scale in which 0 = healthy plants and 100% = plants completely covered by SBDM symptoms/signs. The interaction between the Mg treatments and the fungicide treatments did not significantly affect SBDM severity (**a**). Each Mg concentration (**b**) and fungicide treatment (**c**) followed by a different letter is significantly different from the others, according to two-way ANOVA with Tukey's HSD test. (**d**) The relationship between shoot concentrations of Mg and SBDM severity at 92 days after planting. Default significance levels were set at $\alpha = 0.05$. Bar = SE. The best-fit regression formulas are presented. The Pearson regression values ($r$) are presented along with significance levels ($P$).

Shoot yields of the Mg treatments ranged between 1.165 and 1.388 Kg/m$^2$, with no significant differences between the treatments. Similarly, yields of the Mg × fungicide treatments ranged between 1.753 and 1.967 Kg/m$^2$, with no significant differences between the treatments. Yields of the plots treated with fungicide were significantly higher than the yields of the plots that were not sprayed with fungicide.

## 4. Discussion

These experiments revealed that Ca and Mg applied in the fertigation solution decreased SBDM severity in sweet basil leaves. In contrast, the addition of K to the fertigation

solution increased SBDM severity. The foliar application of Ca and Mg suppressed the disease and foliar-applied K contributed to disease reduction under commercial conditions.

N can also affect the development of SBDM. In a previous greenhouse study, we found that higher levels of N in the fertigation solution led to more severe SBDM and that the use of a fertigation solution in which $NH_4^+$ accounted for less than 10% of the total N was also associated with more severe SBDM (unpublished). To prevent interactions between the effects of the total N and the proportion of N that was $NH_4^+$ and the effects of each of the cations examined in this work, in the experiments reported here, we kept the concentration of the supplied N fairly high: 5.7 and 4.3 mM in the pot and greenhouse experiments, respectively.

The effect of K in this pathosystem may differ from its effects in other pathosystems. The addition of K to a low-nitrate fertilizer has been shown to reduce cucumber fruit gray mold (*Botrytis cinerea*) and stem infections [35]. In addition, foliar-applied $KNO_3$ has been shown to reduce the incidence and severity of Alternaria leaf blight (*Alternaria macrospora* and *A. alternata*) in cotton, as well as leaf-shedding in that crop [36]. Similar to the present work, adding Ca to irrigation water has been shown to reduce gray mold on cucumber fruits and stems. Ca also reduces pepper and eggplant gray mold [35]. In soybean (*Glycine max*), K, Ca, Mg, S and Fe all induce resistance to *Fusarium oxysporum* infection. That effect was observed with moderate concentrations of K and Ca, but not with high concentrations of those nutrients [37]. Another study reported that crown and root rot of tomato caused by *F. oxysporum* f. sp. *radicis-lycopersici* is not affected by $MgSO_4$ but is reduced by $Ca(NO_3)_2$; the authors of that work related the latter effect to the level of nitrate [38].

There have also been previous reports on the effects of nutritional elements on downy mildews. Interestingly, K enrichment has been shown to reduce the natural incidence of downy mildew (*Pseudoperonospora cubensis*) [36]. Seed treatment with 90 mM $CaCl_2$ was shown to suppress downy mildew (*Sclerospora graminicola*) of pearl millet and reduce the pathogen biomass in the treated plants [21]. The application of $KSO_4$ has been shown to reduce the incidence of downy mildew (*Plasmopara viticola*) incidence on grape (*Vitis vinifera*) leaves. Increased K content in grapevine petioles increases the constitutive and post inflectional accumulation of total phenols and phenolic acids such as o-coumaric acid, p-coumaric acid with amplified phenylalanine ammonia-lyase activity in leaves and increased disease resistance [39].

In the present research, the concentrations of Ca and Mg in the shoots were negatively correlated with disease severity. Surprisingly, Ca and Mg did not have an additive effect on SBDM control. The Mg concentration in Ca-enriched plants was lower than that observed in the plants that received the lower Ca treatment. It is possible that the deleterious effect of Ca on Mg load in the canopy was the reason that the combination of supplemental Mg and supplemental Ca did not provide superior disease control. There is no evidence in the literature for such an effect of this cations combination on plant disease. Furthermore, the observed suppression of SBDM by Ca and Mg treatments points to a general mode of action that is triggered by either of those cations. The fact that the application of K through the fertigation solution did not provide similar SBDM suppression, but actually increased SBDM severity, suggests that this mode of action is not generally related to all cations but is specific to $Ca^{2+}$ and $Mg^{2+}$.

Similar to our work, application of 4–30 mM $KNO_3$ prior to inoculation was shown to greatly reduce the incidence of Phytophthora stem rot (the oomycete *Phytophthora sojae*) in soybean. The extent of that disease reduction was related to the increased K concentration in the plants, particularly the accumulation of K in the cortex layer of the plants [40]. In another study, downy mildew (*S. graminicola*) of pearl millet was alleviated by the application of dipotassium hydrogen phosphate under experimental and commercial greenhouse conditions. However, in contrast to our findings, the disease suppression in that system was related to the phosphate component [41]. In the present work, we can conclude that the observed effect was essentially associated with the cations. Contrasting results have been reported for KCl fertilization, which reduced the severity of wheat leaf

rust (*Puccinia triticina*), but that response may have been partially related to the chloride in the KCl fertilizer, as suggested by the authors of that work [42]. In another pathosystem (cucumber downy mildew—*P. cubensis*), increasing the osmotic pressure of a nutrient solution was reported to suppress the expansion of lesion area more effectively than increased leaf concentrations of K, P, Ca and Mg [43].

The choice of anion to be paired with the cations that were used was an important question during the current research. As mentioned above, Cl did not have a clear effect in the early pot work with the cation sprays. In the pot experiments, Cl and $SO_4$ K salts had similar effects on SBDM. Under greenhouse conditions, fertigation-applied $MgCl_2$ suppressed SBDM somewhat more effectively than fertigation-applied $MgSO_4$. $Cl^-$ was the anion of choice in other greenhouse experiments, as well.

Mg and Ca supplied through the fertigation solution suppressed the disease, but in cases of severe epidemics of downy mildew, further disease reduction is necessary. Better SBDM suppression was obtained when the application of those cations through the fertigation solution was paired with the foliar application of K. However, SF calculations revealed significant synergism for only some of the foliar K × cation-supplemented fertigation treatments. A more pronounced improvement in disease control was obtained by combining Mg fertigation with a foliar fungicide treatment; some of the Mg × fungicide treatments provided synergetic disease control. The Mg treatment allows the use of a reduced amount of fungicide, which may help to limit the level of fungicide residue on the harvested branches. A similar benefit from the combination of plant nutrition and fungicides has been observed in the context of other diseases of sweet basil (i.e., gray mold (*B. cinerea*) and white mold (*S. sclerotiorum*)) [10,11].

## 5. Conclusions

The severity of SBDM can be reduced by adding Ca and Mg to the fertigation solution. Spray applications of $K^+$, $Ca^{2+}$ and $Mg^{2+}$ can also affect disease severity. The positive effects of these cations can assist with management of the disease, alongside chemical fungicides whose permissible use is limited. The mode of action of these cations in disease suppression merits further research. Such future research needs to address the possibility of host plant response to the cations and pathogen, including upregulation of induced resistance pathways.

**Author Contributions:** Methodology, Y.E., U.Y. and Z.K.; formal analysis, Z.N., Z.K. and Y.E.; investigation, Z.N., D.R.-D., Z.K., U.Y. and Y.E.; writing—original draft preparation, Y.E. and U.Y.; supervision, Z.K. and Y.E.; project administration, Y.E., U.Y., Z.K. and D.R.-D.; funding acquisition, Y.E., U.Y. and Z.K. All authors have read and agreed to the published version of the manuscript.

**Funding:** This research was funded by Israeli Chief Scientist Ministry of Agricultural and Rural Development (grant #132–1702).

**Institutional Review Board Statement:** Not applicable.

**Informed Consent Statement:** Not applicable.

**Data Availability Statement:** The data that support the findings of this study are available from the corresponding author upon reasonable request.

**Acknowledgments:** We thank the following for their support and assistance during the course of this work: Fertilizers and Chemicals Ltd. (Haifa, Israel), for preparing and supplying the nutrient solutions used in this study; Ran Shulkhani, Ina Finegold, Menachem Borenshtein and Ben Lahav (Agricultural Research Organization); and Ziva Gilad, Efraim Tzipelevitch and Achiam Meir (Tzevi Experimental Station).

**Conflicts of Interest:** The authors do not have any conflict of interest.

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
