# Peer review of "Effects of Calcium, Magnesium and Potassium on Sweet Basil Downy Mildew (Peronospora belbahrii)"

_agronomy, doi:10.3390/agronomy11040688_

Round 1

Reviewer 1 Report

This article is interesting, useful and well prepared.

Introduction it is in line with the Instructions for the Authors. The methodology is also corresponding to the experimental part points of interest. Results are representative and comprehensive.

Discussion is also appropriate, but a more precise direction of future research should add more value.

References are in line with the scientific demonstration of the main issues.

This is a carefully done study and the findings are of considerable interest. 

It is an interesting research work and it is recommended to publish it as a first from a series with incoming findings.

Yours sincerely,

Author Response

This article is interesting, useful and well prepared.

YE response: Thank you.

Introduction it is in line with the Instructions for the Authors. The methodology is also corresponding to the experimental part points of interest. Results are representative and comprehensive.

YE response: Thank you.

Discussion is also appropriate, but a more precise direction of future research should add more value.

YE response: I inserted in the conclusions our ideas of " The mode of action of these cations in disease suppression merits further research. …Such future research needs to address the possibility of host plant response to the cations and pathogen, including upregulation of induced resistance pathways."

References are in line with the scientific demonstration of the main issues.

YE response: Thank you.

This is a carefully done study and the findings are of considerable interest. 

YE response: Thank you.

It is an interesting research work and it is recommended to publish it as a first from a series with incoming findings.

YE response: Thank you.

Reviewer 2 Report

This is a well-written paper with well-executed experiments.  I am especially impressed with the clarity of the figures and tables.  There is only one table that is not completely clear, and that is Table 1.  It lists the experiments performed, which is very helpful, but it calls the experiments one thing in the table and another in the text (for example, "B B1" in the table appears to be the "Experiment B-s" described in the text).  It would be very helpful for the text and table designations to be the same for easy reference. 

The "A" and "B" experiments are (I believe) described in the text (page 5) as having each been performed twice, but this is not indicated in Table 1.  It would make Table 1 longer to indicate replicate experiments, but it would be reassuring.  As a reviewer, I have had to misfortune to review manuscripts where the lack of repeated experiments was deliberately obfuscated, and there is no reason for that here.  The greenhouse experiments clearly indicate when replication occurred without undue bulkiness.  It could be as simple as changing the information in "season" to say "twice during year" or adding an extra column for "number of trials".  It looks as if experiment C3 was not replicated? 

Each graph and table could also indicate from how many trials the data derives.  And you could put a line in the "data analysis" section indicating that data was combined from replicate experiments if no interactions were observed.

There are a few minor points in the text, which I have difficulty describing because I don't know how to indicate line numbers on the pdf I received.

Page 2: the description of potassium deficiency includes the phrase "retarded roots (Yigal Elad, personal information)."  I think because the word "development" was used earlier in the sentence you are reluctant to use it again for "root development", but "retarded roots" is not clear: is the initiation of roots delayed?  Are they shorter than normal?  Sparser? That needs to be clarified.  "personal information" would be "personal communication" if a person not an author of the paper were providing the information, but since this is an author of the paper, perhaps "unpublished data" would be better, here and on page 3?

On page 15 it might be clearer to use the experiment designation from Table 1 to explain where the leaves used in the germination experiment came from.

Author Response

This is a well-written paper with well-executed experiments.  I am especially impressed with the clarity of the figures and tables.  There is only one table that is not completely clear, and that is Table 1.  It lists the experiments performed, which is very helpful, but it calls the experiments one thing in the table and another in the text (for example, "B B1" in the table appears to be the "Experiment B-s" described in the text).  It would be very helpful for the text and table designations to be the same for easy reference. 

YE response: Experiments codes were corrected in table 1 and the corresponding subtitles in the M&M and results section.

The "A" and "B" experiments are (I believe) described in the text (page 5) as having each been performed twice, but this is not indicated in Table 1.  It would make Table 1 longer to indicate replicate experiments, but it would be reassuring.  As a reviewer, I have had to misfortune to review manuscripts where the lack of repeated experiments was deliberately obfuscated, and there is no reason for that here.  The greenhouse experiments clearly indicate when replication occurred without undue bulkiness.  It could be as simple as changing the information in "season" to say "twice during year" or adding an extra column for "number of trials".  It looks as if experiment C3 was not replicated? 

YE response: The reviewer is right. The number of repetitions of experiments A and b is now mentioned in the table and in the descriptions of A and B experiments. Experiments C were indeed repeated as autumn and spring crops, nevertheless since the growing conditions are different (temperatures, RH, canopy wetness) and the take-up of nutritional elements may be different, we suggest not to group them in the detailed description of the experiments. Nevertheless, spring and autumn experiments are grouped in Table 1. C3 indeed was performed in one season only but it repeats treatments from the previous 4 C experiments.

Each graph and table could also indicate from how many trials the data derives.  And you could put a line in the "data analysis" section indicating that data was combined from replicate experiments if no interactions were observed.

YE response: The information regarding the number of trials and reps is inserted in the results section and the indication of experiments data combining is inserted in the data analysis M&M section.

There are a few minor points in the text, which I have difficulty describing because I don't know how to indicate line numbers on the pdf I received.

Page 2: the description of potassium deficiency includes the phrase "retarded roots (Yigal Elad, personal information)."  I think because the word "development" was used earlier in the sentence you are reluctant to use it again for "root development", but "retarded roots" is not clear: is the initiation of roots delayed?  Are they shorter than normal?  Sparser? That needs to be clarified.  "personal information" would be "personal communication" if a person not an author of the paper were providing the information, but since this is an author of the paper, perhaps "unpublished data" would be better, here and on page 3?

YE response: Corrected.

On page 15 it might be clearer to use the experiment designation from Table 1 to explain where the leaves used in the germination experiment came from.

YE response: Separate experiments were conducted for the germination evaluation. The germination experiments are not mentioned in Table 1 since the only disease suppression experiments are included in the Table.